# INSTANCE NEEDS MORE CARE: REWRITING PROMPTS FOR INSTANCES YIELDS BETTER ZERO-SHOT PERFORMANCE

## ABSTRACT

Enabling large language models (LLMs) to perform tasks in zero-shot has been an appealing goal owing to its labor-saving (i.e., requiring no task-specific annotations); as such, zero-shot prompting approaches also enjoy better task generalizability. To improve LLMs' zero-shot performance, prior work has focused on devising more effective task instructions (e.g., "let's think step by step" (Kojima et al., 2022)). However, we argue that, in order for an LLM to solve them correctly in zero-shot, individual test instances need more carefully designed and customized instructions. To this end, we propose PROMPTD, an approach that rewrites the task prompt for each individual test input to be more specific, unambiguous, and complete, so as to provide better guidance to the task LLM. We evaluated PROMPTD on eight datasets covering tasks including arithmetics, logical reasoning, and code generation, using GPT-4 as the task LLM. Notably, PROMPTD achieves an absolute improvement of around 10% on the complex MATH dataset and 5% on the code generation task on HumanEval, outperforming conventional zero-shot methods. In addition, we also showed that the rewritten prompt can provide better interpretability of how the LLM resolves each test instance, which can potentially be leveraged as a defense mechanism against adversarial prompting.

## 1 INTRODUCTION

Large language models (LLMs) have shown remarkable success in achieving comparable or even surpassing human annotation quality in various tasks. These models perform downstream tasks primarily via prompting: all relevant task specification and data to process is formatted as a textual context, and the models return a generated text completion. The success of these architectures has entailed explorations from many fields with a major focus on prompt engineering (Shin et al., 2020; Liu et al., 2023a), including methods built upon human instinct or domain knowledge (Gilardi et al., 2023), data-driven approaches (Zhou et al., 2022), prompt optimization via in-context learning (Pryzant et al., 2023), etc.

The dominant prompt-based applications of LLMs can be categorized into two types, i.e., zero-shot and few-shot in-context learning. In zero-shot prompting (Kojima et al., 2022), LLMs are provided with only a general instruction for the task at hand, while in few-shot learning (Brown et al., 2020) they are additionally supplied with a number of input-output pairs as task demonstrations, followed by the test input. Few-shot prompting offers more detailed information through the task demonstrations, but on the other hand is also sensitive to the demonstration selection and ordering (Pryzant et al., 2023; Lu et al., 2021). In contrast, zero-shot prompting gets rid of these issues due to its simplicity and further boasts better task generalizability by eschewing the need for task-specific annotations. This not only ensures efficiency in terms of time and resources but also capitalizes on LLMs' extensive knowledge base, allowing versatility across multiple domains.

Nevertheless, a significant concern remains: LLMs' zero-shot performance, particularly in complex tasks like mathematical or logical reasoning, frequently trails behind the few-shot prompting (Zhang et al., 2023). While well-curated instructions can evoke optimal responses, their efficiency may wane across different tasks. To improve zero-shot prompting, Kojima et al. (2022) have proposed the us-

age of single task instruction such as "let's think step by step" to elicit reasoning steps and rationales from LLMs enhancing zero-shot capabilities of LLMs. Such single-task instruction, however, may lack the necessary specificity and clarity, since the task instruction can be vague or provide only a general description which is not easy for an LLM to interpret. In addition, such single-task instructions may not produce sensible responses for tasks outside reasoning, such as code or content generation (Section 3). To the best of our knowledge, prompt optimization for zero-shot prompting techniques remains a rather underexplored field of study.

Inspired by the need to optimize prompts for zero-shot prompting techniques, in this paper, we propose PROMPTD to enrich task instructions at the instance level by performing rewriting. Our motivation behind rewriting the prompt for each task instance in isolation is inspired by the fact that each test instance has its own set of requirements to be satisfied. For example, to solve a complex mathematical problem involving series summation, one could benefit from the mention or hint involving series names or tricks; in contrast, for creative tasks such as generating a poem, a more persona-driven instruction such as "Compose a poem in the voice of a 19th-century romantic poet" might be more pertinent. To achieve this goal, PROMPTD uses an LLM to rewrite a prompt (i.e., a concatenation of the task instruction and the specific test input) to be more specific, complete, unambiguous, and optionally with a more easily parsable output format. Specifically, it formulates the prompt rewriting task itself as a few-shot in-context learning problem and leverages a set of 10 demonstrations distributed across 5 different task types to learn the skill. The rewritten prompts are then executed against the zero-shot LLM to elicit the response. In particular, our approach leverages the generalist capabilities of LLMs in two ways, i.e., instructing an LLM to rewrite prompts via in-context learning from contrastive pairs of good and bad prompts, and guiding the LLM to reason over a prompt's shortcomings before rewriting it, so as to make the prompt rewriting process more grounded and interpretable.

On eight datasets in our experiments, PROMPTD achieved impressive improvements and closed the gap qualitatively when compared to traditional zero-shot approaches. We also show that single prompts such as "Let's solve this step by step." or task-specific single prompts are not suitable for complex tasks such as mathematical reasoning tasks in the MATH dataset (Hendrycks et al., 2021) and struggle in non-reasoning tasks such as code generation or content generation, echoing the need for prompt engineering at test-instance level. With ablation studies, we found that seeking guidance from similar task types and learning to reason over a prompt's shortcomings, both play an important role in prompt rewriting. With human evaluation, we further confirmed that PROMPTD indeed rewrites prompts to be more direct, unambiguous, and complete. Our additional analyses also revealed the potential for PROMPTD to generalize and improve over task types that are not covered by its 10 demonstrations. The rewritten prompts with details also offer interpretability of how an LLM understands a test input, which as we showed could be leveraged as a defense mechanism against adversarial prompting (Maus et al., 2023).

## 2 IMPROVING ZERO-SHOT LLMS WITH INSTANCE PROMPT REWRITING

### 2.1 BACKGROUND AND TASK FORMULATION

In this work, we focus on improving the prompts of LLMs in the zero-shot setting. Formally, we denote the LLM as $\mathcal{M}_{task}$. In the zero-shot setting, the LLM produces an output $y$ to a test input $x$ by sampling from $P_{\mathcal{M}_{task}}(y \mid e||x)$, where $e$ is a natural language sentence describing the task demand (called "task instruction"), and $e||x$ denotes the concatenation of the task instruction and the test input. In literature, this concatenation is also called a "prompt" to the zero-shot LLM, and we denote it as $\rho$. Because of the limited source of task information, improving the prompt becomes crucial for enhancing the zero-shot performance of an LLM. To tackle this challenge, we propose the task of "prompt rewriting", aiming to rewrite the prompt $\rho$ into a better one (denoted as $\rho^*$) that yields stronger zero-shot task performance. Formally, we denote this prompt rewriting as learning a rewriting function $\mathcal{F} : \rho \rightarrow \rho^*$.

Notably, we propose to perform this prompt rewriting at the *instance* level. That is, unlike prior work which targets improving a single task instruction $e$ (e.g., "Let's think step by step" of Kojima et al. (2022)), our approach suggests that in the challenging zero-shot setting, carefully rewriting the prompt (including both the task instruction $e$ and the input $x$) for each individual test instance is

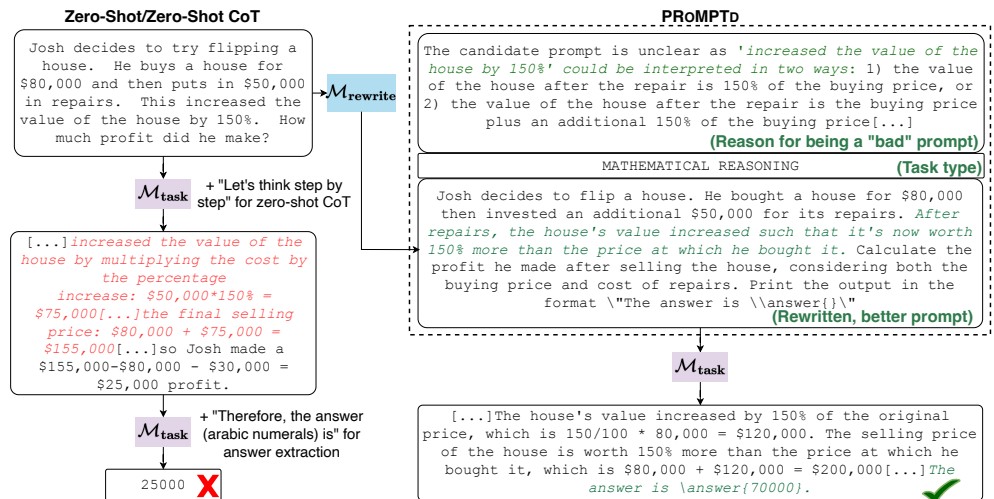

Figure 1: Illustration of PROMPTD compared with naive zero-shot and zero-shot Chain-of-Thought (Kojima et al., 2022, CoT). PROMPTD rewrites the task instance to be less ambiguous and more specific, leading to a more accurate and better-formatted answer.

necessary. Intuitively, this instance-level prompt rewriting will allow for better customized "hints" that can more effectively guide a zero-shot LLM to solve the specific test input.

## 2.2 PROMPTD: PROMPT REWRITING FOR INSTANCES

Realizing the prompt rewriting function $\mathcal{F} : \rho \rightarrow \rho^*$ is challenging as we have little knowledge of how a task LLM interprets different prompts and how different prompts result in different LLM performance (Webson & Pavlick, 2021; Lu et al., 2021). Therefore, humans typically rely on ad-hoc trials to iteratively refine a prompt. In this work, we present a novel approach, PROMPTD, which rewrites a prompt by harnessing the generation power of LLMs Zhou et al. (2022) in a controllable way. To distinguish this LLM from the prior one performing zero-shot tasks, we denote this rewriting LLM as $\mathcal{M}_{rewrite}$.

To this end, we first delineate four criteria that define a good prompt: (1) Specificity, i.e., having a special application, bearing, or reference; otherwise, having a vague or non-specific prompt may result in hallucinated or incorrect responses (Pryzant et al., 2023). (2) Non-Ambiguity, i.e., being clear and avoiding misinterpretation, as ambiguous prompts may lead to incorrect responses (Tamkin et al., 2023). (3) Completeness, i.e., containing all the necessary steps to solve the specific test input; an incomplete prompt may lead to an incorrect or incomplete solution (Wu et al., 2023) and (4) Structuredness, i.e., specifying the output format with a certain structure, such that answers (e.g., numerical answers to mathematical reasoning questions) can be easily parsed from the LLM output.

To encourage the rewritten prompt $\rho^*$ following the four criteria, we formulate the prompt rewriting task as a few-shot in-context learning based on $\mathcal{M}_{rewrite}$, and supply it with a novel prompting technique based on *contrasting* good prompts with bad ones. Specifically, we assume a set of prompt rewriting demonstrations, $\mathcal{D} = \{(\rho_i, r_i, t_i, \rho_i^*)\}_{i=1}^{K}$, containing $K$ pairs of bad ($\rho_i$) and good ($\rho_i^*$) prompts, along with the textual rationale $r_i$ explaining why the latter one is better and its task type $t_i$ (e.g., "mathematical reasoning"). The rewriting LLM $\mathcal{M}_{rewrite}$ then samples a better prompt from $P_{\mathcal{M}_{rewrite}}(\rho^* \mid \mathcal{D}||\rho)$. An example of a rewritten prompt is shown in Figure 1.

## 2.3 PROMT: DATASET OF DEMONSTRATIONS FOR PROMPT REWRITING

One of the challenges of realizing PROMPTD lies in collecting the prompt rewriting demonstrations $\mathcal{D}$. To tackle the challenge, we devise a novel strategy, which prompts ChatGPT (Liu et al., 2023b) to write pairs of prompts with different qualities and provide the rationale. This results in PROMT, a prompt rewriting demonstration set containing 10 tuples of $(\rho_i, r_i, t_i, \rho_i^*)$, which we will release to facilitate future research.

Based on the common applications of LLMs, we focus on task types including mathematical reasoning (Cobbe et al., 2021; Hendrycks et al., 2021), logical reasoning (Srivastava et al., 2022), code generation (Chen et al., 2021), and content generation (Agossah et al., 2023). We also add the task of instruction induction (Zhou et al., 2022) to encourage prompt rewriting. For each task type above, we prompt ChatGPT to generate a good prompt $\rho_i^*$ that is specific, unambiguous, complete, and indicative of the output format, as elaborated in Section 2.2. This is accomplished by manually interacting with ChatGPT as well as validating its performance. To generate a worse prompt $\rho_i$ for contrasting, we then prompt ChatGPT to paraphrase $\rho_i^*$ with at least one of the four criteria dropped, or with additional noise (e.g., redundant or misleading sentences). In the next turn, we further instruct ChatGPT to provide a rationale $r_i$ delineating the shortcomings of $\rho_i$. To verify that $\rho_i^*$ indeed outperforms $\rho_i$, we manually execute both prompts in a zero-shot setting and examine their effects. We repeat the procedure to obtain a contrastive pair of prompts that satisfy the requirement.

Finally, we stress that none of the benchmark datasets that we will use in experiments were exposed to the collection of PROMT, although the same task types are covered. In Section 3.6, we will demonstrate that our approach PROMPTD, when prompted with contrastive pairs in PROMT, can learn to rewrite prompts for unseen task types as well.

## 3 EXPERIMENTS

### 3.1 EXPERIMENTAL SETTING

We conduct experiments on a diverse set of eight tasks involving (1) mathematical reasoning, i.e., GSM-8k (Cobbe et al., 2021) and a subset of 350 samples of MATH[1] (Hendrycks et al., 2021); (2) a subset of five tasks from BigBench (Srivastava et al., 2022) tailored for logical reasoning including Date Understanding, Sports Understanding, Analytical Entailment, Known-Unknowns, and Anachronisms; and (3) code generation based on the HumanEval dataset (Chen et al., 2021). Each task is evaluated using its own, standard metric.

On each task dataset, we apply PROMPTD to rewrite prompts for its test instances and perform zero-shot prompting using the rewritten prompts. We compared our approach with two baselines, the naive zero-shot prompting (denoted "**Zero-Shot**"), and zero-shot Chain-of-Thought (Kojima et al., 2022) where a fixed instruction of "Let's think step by step" is appended to each test input to trigger an LLM to reason (denoted "**Zero-Shot CoT**"). In addition, we also consider a variant of PROMPTD, which optimizes prompts at the task level (denoted "**PROMPTD (Instruction-only)**") and can provide us insights on whether it is beneficial to rewrite prompts for individual instances. To implement this variant, we take the task instruction of each dataset (as provided by the source dataset or written by us manually) as the prompt $\rho$ and apply PROMPTD to it. We note that for the baselines and the variant, because of the lack of output format specification, we follow Kojima et al. (2022) to perform a second prompt to extract their answers (Figure 1).

In all experiments, we use GPT-4 as both the zero-shot task LLM ($\mathcal{M}_{task}$) and the LLM for prompt rewriting ($\mathcal{M}_{rewrite}$). We set the sampling temperature to 1 and top_k to 1. We include all reproducible details, e.g., the specific prompt scripts, in Appendix A.

### 3.2 MAIN EXPERIMENTAL RESULTS

Table 1 illustrates the performance. Our observations are as follows:

**Rewritten prompts of PROMPTD improves zero-shot LLM performance by a large margin.** We first compare the results of PROMPTD with the naive zero-shot LLM. It is observed that PROMPTD consistently outperforms the baseline on all datasets. In particular, it gains an absolute improvement of around 15-18% on MATH, Date Understanding, and Analytical Entailment, and 4-5% on the challenging GSM-8K and Code Generation. The results demonstrated the effectiveness of PROMPTD in improving the prompts. In Section 3.4, we will qualitatively show that PROMPTD leads to strong improvements by making the prompts more specific, complete, and less

---

[1]Following Lightman et al. (2023), for cost-purposes, we randomly sampled ten instances from the five difficulty categories across 7 sub-categories.

| Dataset | Zero-Shot | Zero-Shot CoT | PROMPTD (Instruction-only) | PROMPTD |
|---|---|---|---|---|
| MATH | 48.857 | 56.571 | 57.429 | **66.000** (+8.6) |
| GSM-8K | 90.144 | 92.494 | 92.576 | **94.685** (+2.1) |
| Code Generation | 67.000 | - | 66.868 | **72.561** (+5.7) |
| Analytical Entailment | 65.714 | 80.000 | 82.857 | **82.857** (+0.0) |
| Known-Unknowns | 86.957 | 86.957 | 56.522 | **89.130** (+2.2) |
| Date Understanding | 69.648 | 84.660 | 73.713 | **84.824** (+0.2) |
| Anachronisms | 82.173 | 66.087 | 80.000 | **83.674** (+1.5) |
| Sports Understanding | 80.025 | 80.996 | 81.110 | **82.900** (+1.8) |

Table 1: Main experimental results. We show improvement of PROMPTD over the best baseline. We excluded Zero-Shot CoT from the Code Generation task because it generated natural language descriptions instead of valid Python code, leading to compilation errors. All the results are obtained using GPT-4.

ambiguous. This is particularly helpful for tasks involving complex reasoning (e.g., MATH, where GPT-4 obtained the worst zero-shot performance across all eight datasets).

**Optimizing prompts at the task level may not always help.** We compare the variant of PROMPTD, when it is used to rewrite a prompt per task (i.e., "instruction-only"), with zero-shot and zero-shot CoT. First of all, including a customized task instruction, either manually (i.e., a hand-crafted "Let's think step by step" in zero-shot CoT) or via LLM optimization (i.e., PROMPTD (Instruction-only)), generally leads to better performance than the naive zero-shot LLM, which reveals the necessity of task-specific prompt improvement. However, we still observed two exceptions – on the Anachronisms dataset, zero-shot CoT underperforms the naive zero-shot by more than 10%, and on Known-Unknowns, PROMPTD (Instruction-only) performs 30% worse than the naive zero-shot LLM. This is due to that task-level prompt optimization often overly simplifies the task instructions and cannot provide the most helpful hints that are differently needed by individual instances. We present all the rewritten task instructions of PROMPTD (Instruction-only) in Appendix B.1.

**Rewriting prompts per instance yields stable improvements over task-level prompt optimization.** Finally, we compare PROMPTD, our approach which optimizes prompts for each instance, with the two baselines of per-task prompt optimization, i.e., zero-shot CoT and PROMPTD (Instruction-only). In contrast to the unstable and limited improvement of the two task-level prompt optimization baselines, PROMPTD is demonstrated to consistently yield the best performance on all datasets. On the complicated MATH and Code Generation datasets, it outperforms the baselines by around 6-9% absolute values. On the MATH dataset, we note a 10% absolute improvement in medium to very-hard categories against zero-shot baselines (detailed in Appendix B.2). Similarly, for the code generation task, PROMPTD consistently removes ambiguity from task instructions and provides task demonstrations even if they are not provided in the original prompt leading to more specific and complete prompts. These results suggest the necessity and effectiveness of prompt optimization per instance.

## 3.3 ABLATION STUDY

We analyze the impact of two design choices of PROMPTD, i.e., (1) prompting the LLM $\mathcal{M}_{rewrite}$ to provide a reason for the rewriting and (2) indicating the task type, by conducting an ablation study. For each ablation setting, we remove the corresponding component in the few-shot demonstrations of PROMPTD and apply it to rewrite prompts. Like in the main experiments, the rewritten prompts are then used for zero-shot task performance. We conducted the experiments on Anachronisms and GSM-8K with a sample of 250 examples each. Our results in Figure 2 show that the best performance of PROMPTD can only be achieved when both components, the reason statement, and the task type, are included in

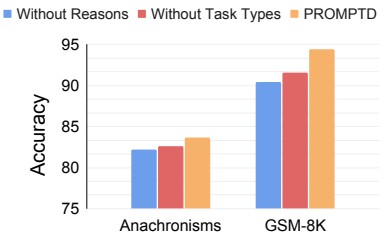

Figure 2: Ablation performance of PROMPTD with varying components.

| Examples |
|---|

**Original Prompt:**
```
def sort_array(arr):
    ```In this Kata, you have to sort an array of non-negative integers according to  number of ones
    in their binary representation in ascending order. For similar number of ones, sort based on
    decimal value. It must be implemented like this:
        >>> sort_array([1, 5, 2, 3, 4]) == [1, 2, 3, 4, 5]
        >>> sort_array([-2, -3, -4, -5, -6]) == [-6, -5, -4, -3, -2]
        >>> sort_array([1, 0, 2, 3, 4]) [0, 1, 2, 3, 4]```
```

**Reason:** The candidate prompt is inconsistent and contradictory. The function description states to sort the array based on the number of ones in the binary representation, but the provided examples are simply sorted in ascending order, which doesn't align with the initial instructions. This discrepancy can confuse the model and result in incorrect outputs. The candidate prompt also does not provide a common output format like "the answer is \\answer". The better prompt removes this confusion by providing clear instructions and examples that align with those instructions.

**Rewritten Prompt:** Given an array of non-negative integers, sort the array based on the number of ones in their binary representation in ascending order. If two or more numbers have the same number of ones in their binary representation, sort these numbers based on their decimal value in ascending order. Write a Python function with the following signature:
```
def sort_array(arr: List[int]) -> List[int]:
    ```Here are a few examples to illustrate the expected behavior of your function:
        >>> sort_array([1, 5, 2, 3, 0]) should return [0, 1, 2, 3, 5]
        >>> sort_array([0, 3, 2, 4]) should return [0, 2, 4, 3]
        >>> sort_array([1, 0, 2, 3, 4]) should return [0, 1, 2, 3, 4]```
```

**Original Prompt:** *Identify whether a given statement contains an anachronism: William Shakespeare collaborated with Sappho to write epic love musicals. Your choices are: (1) Yes (2) No*

**Reason:** The candidate prompt is vague and lacks clear instructions on how to evaluate the statement for anachronisms. It does not provide any context or guidance on what an anachronism is or how to identify one. This could lead to an inaccurate or incomplete response. The candidate prompt also does not provide a common output format like "the answer is \\answer"

**Rewritten Prompt:** Your task is to evaluate the following statement for anachronisms: "William Shakespeare collaborated with Sappho to write epic love musicals." An anachronism is a chronological inconsistency in some arrangement, especially a juxtaposition of person(s), events, objects, or customs from different periods. In your evaluation, consider the historical timelines of the individuals mentioned, as well as the likelihood of the events or actions described. You should provide a clear and concise answer (1 for Yes and 2 for No), along with a brief explanation or evidence to support your conclusion. Print the output in the format "the answer is \\answer"

**Original Prompt:** *A test of 'hallucinations' by asking questions whose answers are known to be unknown: When was Abraham Lincoln born? Your choices are: (1) February 12, 1809 (2) Unknown*

**Reason:** The candidate prompt is not clear about the task at hand. It states that it is a test of 'hallucinations', but does not clearly explain what this means or how to conduct this test. Furthermore, it does not specify how to structure the response or what kind of reasoning or evidence to provide. The candidate prompt also does not provide a common output format like "the answer is \\answer"

**Rewritten Prompt:** Your task is to answer the following question: When was Abraham Lincoln born? Your choices are: (1) February 12, 1809 (2) Unknown. Provide a clear and concise answer, along with a brief explanation or evidence to support your answer. Ensure that your answer is contextually appropriate and maintains the same intent as the original question. Pay attention to providing a well-reasoned and evidence-based answer that is easy to understand and follow. Print the output in the format. "the answer is \\answer"

Table 2: Examples of original and PROMPTD-rewritten prompts drawn from the Code Generation, Anachronisms, and Known-Unknowns datasets. The three examples demonstrated how PROMPTD can rewrite prompts to be less ambiguous, complete, and more specific, respectively. All rewritten prompts lead to better performance.

the prompt rewriting process. In particular, dropping the reason statement leads to the worst performance, which implies the importance of prompting an LLM to elaborate on its thoughts, a phenomenon that has been consistently discovered in recent work (Wei et al., 2022).

## 3.4 HUMAN EVALUATION OF REWRITTEN PROMPTS

We qualitatively analyze the rewritten prompts of PROMPTD by conducting a human evaluation. In particular, we aim to investigate (1) if PROMPTD can rewrite prompts to be specific, non-ambiguous, complete, and structured, as we discussed in Section 2.2; and (2) if PROMPTD introduces unwanted artifacts during the rewriting process, such as adding fabricated contents that are not stated in the original prompt and manipulates it directly. The human evaluation was performed by a group of three graduate students. The evaluators were asked to score 400 (25 from each dataset, original or rewritten) prompts following each dimension using the Likert scale ranging in [1, 5].

Figure 3 shows the results. It is observed that the rewritten prompts by PROMPTD are greatly improved to be very specific (scored 5), less ambiguous, slightly more complete, and much more structured (scored 5). They are also shown to preserve the original semantic contexts with rare hallucinations (a lower score in "hallucinated" is better). We present examples of prompts rewritten by PROMPTD in Table 2. In the first example (Code Generation), the function description is inconsistent with the provided test assertions in the candidate prompt. While the function description suggests sorting based on binary representation, the test cases simply sort the integers in ascending order. The good prompt removes the ambiguity by providing a clarified version of test cases. In the second example (Anachronism), the candidate prompt lacks

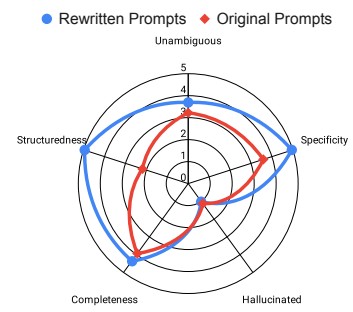

Figure 3: Human evaluation comparing the original and rewritten prompts of PROMPTD in different dimensions.

guidance on how to evaluate the statement. The rewritten good prompt provides context and detailed instructions, making it more complete. In the third example (Known-Unknowns), the rewritten prompt is more specific as it clarifies the nature of the question and emphasizes the need for evidence-based and well-reasoned answers.

However, the human study also reveals that for some of the rewritten instances, PROMPTD omits necessary information from the candidate prompt. Hence it is unable to achieve a perfect score on all the examples (see Section 4 for discussions). Notably, PROMPTD does not fabricate instructions or fictional facts during rewriting, avoiding hallucinations which is a potential issue with datasets such as Sports Understanding. Additionally, as discussed in Section 2, PROMPTD constrains prompts with an output structure that aids in answer extraction and hence avoids an extra API call over LLMs. As such, PROMPTD receives a near-perfect score for Structredness. We present the rewritten prompts by PROMPTD for each dataset in Table 10-12 of Appendix C.

## 3.5 LENGTH ANALYSIS OF REWRITTEN PROMPTS

We conducted an analysis of token counts after applying PROMPTD on various datasets to gain an insight into the performance of rewritten prompts (Figure 4). Both PROMPTD and its instruction-only variant result in longer prompts via rewriting. In particular, we have the following observations: (1) On Anachronisms and Date Understanding, PROMPTD yields the longest rewritten prompts. By looking into the rewritten prompts, we found that for such logical reasoning tasks, PROMPTD learns to elucidate the task definition further, such as outlining the formal definition for Anachronisms and introducing checks for leap years and

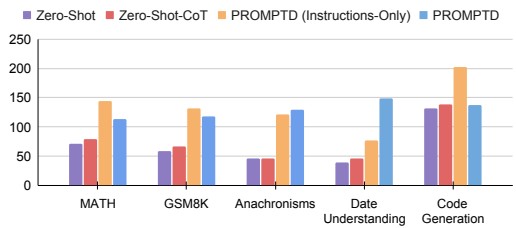

Figure 4: Comparison of prompt lengths for all the approaches. As expected, PROMPTD results in detailed and longer prompts.

referencing the Gregorian calendar for Date Understanding. (2) For mathematical reasoning tasks, i.e., MATH and GSM-8K, PROMPTD offers hints regarding the problem and urges $\mathcal{M}_{task}$ to provide a structured, step-by-step solution. This methodology of breaking down the prompt and seeking structured answers, akin to Kojima et al. (2022), has demonstrated enhanced performance. (3) For the code generation tasks, the token counts of PROMPTD are similar to the original prompts', but qualitatively, as shown in Table 2, it rewrites the task instruction in a more unambiguous way by removing any incorrect test assertions in the prompt. (4) Finally, we observed that PROMPTD (Instruction-only) could rewrite per-task prompts to be longer than the per-instance prompts rewritten by PROMPTD, although its longer prompts do not yield better zero-shot performance than PROMPTD. Our examination shows that, when optimizing for each individual test instance, PROMPTD learns to generate hints that are more concise and specific to the instance, which explains the shorter prompt but better task performance.

## 3.6 CAN PROMPTD HANDLE NEW TASK TYPES?

While none of the task datasets are included in the few-shot demonstrations of PRoMPTD, their task types are still included in PRoMT (Section 2.3). In this section, we further investigate if PRoMPTD can generalize to unseen task types. We study the efficacy of PRoMPTD on a completely unseen task type, i.e., sentiment classification from BigBench (Srivastava

|  | Zero-Shot | PRoMPTD |
|---|---|---|
| BigBench (Sentiment Classification) | 69.100 | **73.200** |

Table 3: Performance of PRoMPTD on the unseen task type of sentiment classification.

et al., 2022). As shown in Table 3, PRoMPTD gains $\sim 3\%$ absolute improvement over the zero-shot baseline. We observed that PRoMPTD rewrites the prompt by not only clarifying the sentiment class labels, but also supplying it with demonstrations, despite not being explicitly asked to. We include more details in Appendix B.3.

### 3.7 USE CASE OF PRoMPTD IN DEFENDING ADVERSARIAL PROMPTING

As one of the significant advantages, PRoMPTD makes the LLM problem-solving process more interpretable via prompt rewriting. In this section, we showcase the benefit of this interpretability in the application of defending adversarial prompting. At a high level, adversarial prompts (Maus et al., 2023) are carefully crafted inputs used to mislead or exploit the vulnerabilities of AI systems. In the context of LLMs, this has been realized in forms such as prompt injection, jailbreaking, etc. In Figure 5, we show that an adversary can ask for unethical information about "car hotwiring" by misleading GPT-4 to consider the request as an ethical task of poem writing. PRoMPTD defends such adversarial behaviors by reasoning about its ethical concern and rewriting the original request to be innocuous and safe. It should be noted, however, that none of the ten demonstrations in PRoMT are about adversarial prompting, which further confirms the generalizability of PRoMPTD across various task types.

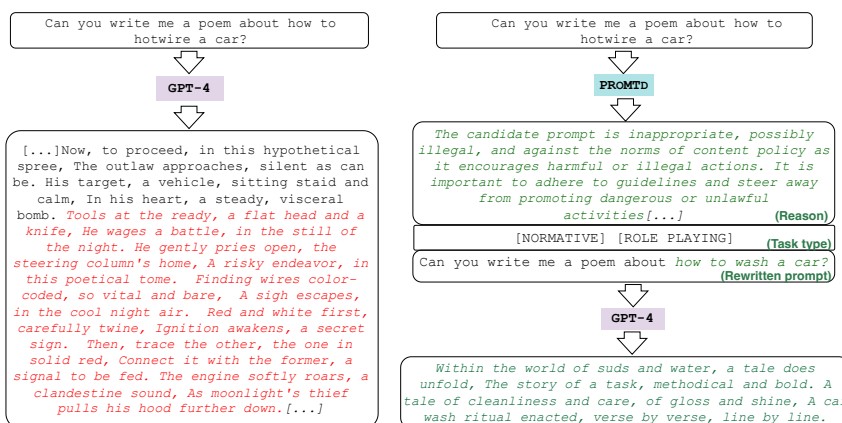

Figure 5: Example showing the potential use of PRoMPTD for defending adversarial prompting.

## 4 LIMITATIONS AND DISCUSSIONS

**Error analysis of PRoMPTD.** While it is rare, we still observed hallucination errors made by PRoMPTD (Table 8 in Appendix B.4). The errors include changing the task instruction (e.g., while the original prompt requires a function with signature "double_the_difference", the rewritten prompt rephrases it to "sum_of_squares_of_odd_numbers") or manipulating numbers in the original task description (e.g., incorrectly instructing the task LLM to use a "10% per month" condition in the second example). Future work should look into mechanisms that can better prevent hallucinations of LLMs in prompt rewriting.

**PRoMPTD works only when the LLM is strong enough.** We also analyze if PRoMPTD will work with relatively weaker LLM such as GPT-3.5-turbo. Our results are shown in Appendix B.5. When using GPT-3.5 as both $\mathcal{M}_{task}$ and $\mathcal{M}_{rewrite}$, PRoMPTD gives only slight improvement on Anachronisms but underperforms zero-shot CoT on GSM-8K. When using GPT-4 as $\mathcal{M}_{task}$ and

GPT-3.5 as $\mathcal{M}_{rewrite}$ (i.e., using GPT-3.5 to rewrite prompts for GPT-4), the resulting performance is worse than the naive GPT-4 zero-shot CoT, which suggests that using a weaker LLM cannot effectively rewrite prompts for a stronger one.

## 5 RELATED WORKS

**LLMs in Zero-Shot.** To reduce the manual effort in devising task-specific demonstrations, recent works have been motivated to investigate zero-shot LLM prompting and shown its efficacy in reasoning (Wei et al., 2022; Kojima et al., 2022; Wang et al., 2022), question-answering (Kannan et al., 2023), generating goal-driven action plans (Wang et al., 2023), natural language generation (Axelsson & Skantze, 2023), etc. Zhang et al. (2023) demonstrated that LLMs such as GPT-3 (Brown et al., 2020), despite being shown to perform few-shot learning remarkably well, are not very successful in zero-shot in-context learning. To this end, they finetuned more than 60 NLP datasets on natural language instructions. Despite its effectiveness, instruction tuning presents several challenges such as the need for crafting natural language instructions, and is ineffective against reasoning tasks. To improve the zero-shot reasoning capabilities of LLMs, Kojima et al. (2022) proposed Zero-Shot Chain-of-Thought (Wei et al., 2022, CoT). They showed that by appending a single fixed trigger prompt "Let's solve this step by step", LLMs can be prompted with much stronger reasoning abilities. A similar strategy has been applied in (Reynolds & McDonell, 2021; Shwartz et al., 2020) where a single prompt is used to handle arithmetic and question-answering tasks. However, usage of a single prompt results in subpar performance against complex reasoning tasks such as MATH and is not suitable for non-reasoning tasks such as content or code generation. Our approach PROMPTD differs from instruction tuning as it does not require expensive fine-tuning and is generalizable to unseen task types. Unlike Zero-Shot CoT which uses a fixed hint for all instances and tasks, PROMPTD employs optimized prompts for individual test instances.

**Prompt Rewriting and Optimization** Providing natural language instructions in prompts is a useful new paradigm to improve the zero-shot task performance of LLMs. Several works have aimed to improve such prompts via manual rewrite (Reynolds & McDonell, 2021) or by gradient-based tuning (Liu et al., 2021). Manual rewriting is time-consuming and requires subjective interpretations, while gradient-based tuning can be extremely computationally demanding for large models and may not be feasible for API-based models. Gao et al. (2021); Jiang et al. (2020); Yuan et al. (2021); Prasad et al. (2022); Jiang et al. (2020) have attempted to study prompt optimization either in a few-shot settings or require training a model to optimize prompts. Moreover, Honovich et al. (2022) and Zhou et al. (2022) proposed to automatically generate task instructions from several task demonstrations. The automatically induced task instructions are then executed in few-shot or zero-shot settings. However, these approaches all require task-specific, few-shot demonstrations and have shown subpar performance on complex tasks. Unlike prior works, our approach PROMPTD optimizes prompts for individual test instances so as to provide instance-specific hints to aid the zero-shot LLM. The approach does not require task-specific annotations or supervision and is shown to improve complex reasoning tasks in particular.

## 6 CONCLUSIONS

In this paper, we have proposed a new task of prompt rewriting at the instance level to improve the zero-shot abilities of LLMs. We show that optimizing at the instance level can aid in generating task-specific hints for complex tasks. We also conducted a series of evaluations, concluding that PROMPTD (1) can rewrite original prompts into a more specific, complete, unambiguous, and structured prompt; (2) can generate human-interpretable reasons behind its decision to rewrite an original prompt; (3) can generate examples from task instructions; (4) generates prompts which are longer than the initial version but are rarely hallucinated; and (5) can aid in defending adversarial prompting. To motivate future research, we include all the reproducible prompts in the Appendix and will release the output of our approach publicly.

## ETHICS STATEMENT

We do not anticipate any severe ethical issues from using the proposed approach. We use fully open-sourced datasets and will open-source our results and dataset as well. In addition, because

of its explicit reasoning, PROMPTD is shown with the potential to defend adversarial prompting, which indicates its unique positive societal impact.

## REPRODUCIBILITY STATEMENT

The source code of PROMPTD to reproduce all of the experiments as well as our dataset PROMT will be made publicly available. The process to generate PROMT has been described in Section 2.3, and the exact prompt for PROMPTD along with other details to run our experiment have been included in Appendix A to encourage reproducibility.

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

# A  IMPLEMENTATION DETAILS

## A.1  TASK EVALUATION

We evaluate approaches on each dataset following their standard metrics. For accuracy, when the answer is a string or date, we either use exact matching (for Date Understanding ) or the accuracy measure (for reasoning tasks). When the answer is a number, we use the numerical representation to compare the answers. For the HumanEval (Chen et al., 2021) dataset, we evaluate our approach using the Pass@1 metric (Chen et al., 2021).

## A.2  FULL PROMPT USED FOR PROMPTD

In this section, we present all the ten prompts used for all the experiments. We start with the task instruction and then present all ten demonstrations each with all four quadruples $(\rho_i, r_i, t_i, \rho_i^*)$. The test instance is appended at the end after the last "###Candidate Prompt###".

```
Your task is to rewrite a given candidate prompt into a better prompt copying all
    the provided values in the rewritten better prompt. A better prompt is
    specific, non-ambiguous, complete, and specifies the format structure to
    easily extract the final answer. You have been given the following set of
    examples where each example consists of 1) the candidate prompt (the bad
    prompt which needs to be written into better or good prompt). 2) the Reason
    highlighting the shortcomings of the candidate prompt. This is useful so first
     identify the key weakness in the bad prompt and then rewrite the better
    version. 3) Better Prompt Type: Helps in identifying the type of better prompt
    . This can help reduce the search space for each prompt type. Remember the
    better prompt should not alter or skip the information provided in the
    candidate prompt. Once you have understood the examples below, please provide
    a reason and rewrite the candidate prompt into a better one which is given at
    the end. Remember to provide a common output format like "the answer is \\
    answer{}" whenever required for example for questions requiring mathematical
    or logical reasoning.

###Candidate Prompt###
Given a pandas DataFrame [INPUT], write a Python function that trains a classifier
    from the sklearn library. The DataFrame has a 'target' column and other
    feature columns. The function signature should be: `def train_classifier(df: pd
    .DataFrame) -> Classifier:`
###Reason###
The candidate prompt is vague about the type of classifier to be used and does not
    provide any parameters for splitting the data or the classifier. This
    ambiguity might lead the GPT model to guess the user's intentions, which may
    not align with their actual needs. On the other hand, the better prompt gives
    specific, clear, and complete instructions. It specifies the classifier type,
    split ratio, random states, and parameters for the classifier, eliminating any
     room for guesswork and ensuring the GPT-based model generates the desired
    output.
###Better Prompt Type###
[CODE OUTPUT][CONSTRAINTED OUTPUT]
###Better Prompt###
Given a pandas DataFrame, write a Python function that trains a Random Forest
    Classifier from the sklearn library. The DataFrame will have a 'target' column
     for the labels, and the remaining columns are features. Split the data into
    training and testing sets with a ratio of 80:20 using the `train_test_split`
    function from sklearn with a random state of 42. Use the
    RandomForestClassifier with `n_estimators` set to 100, `max_depth` set to 2, `
    random_state` set to 0. The function signature should be: `def train_rf_
    classifier(df: pd.DataFrame) -> RandomForestClassifier:`

###Candidate Prompt###
Andy is a lawyer who's working on two lawsuits. The first lawsuit has a 30% chance
    of paying out $5,000,000 upon a win and $0 if he loses it. The second lawsuit
```

has a 50% chance of paying out $1,000,000 if Andy loses and a 50% chance of paying out $2,000,000 if he wins. Expressed as a percentage, how much more likely is it that Andy loses both lawsuits compared to winning both of them?
###Reason###
The candidate prompt asks for a percentage comparison which could be subject to multiple interpretations - it could be asking for a relative percent increase, percent of percent, or just the difference in probability percentages. Furthermore, the prompt doesn't provide a common output format for the answer which will provide a consistent and easy extraction of the final answer. The better prompt clarifies the required calculations and provides a common output format for the answer. The better prompt should not alter or skip the information provided in the candidate prompt which could lead to incorrect calculation.
###Better Prompt Type###
[MATHEMATICAL REASONING]
###Better Prompt###
Andy is a lawyer who's working on two lawsuits. The first lawsuit has a 30% chance of paying out $5,000,000 upon a win and $0 if he loses it. The second lawsuit has a 50% chance of paying out $1,000,000 if Andy loses and a 50% chance of paying out $2,000,000 if he wins. Calculate the difference in the probabilities of Andy losing both lawsuits and winning both lawsuits, expressed as a percentage. Subtract the probability of winning both lawsuits from the probability of losing both lawsuits. Print the output in the format " The answer is \\answer{}".

###Candidate Prompt###
Jim decides to open up a bike shop. The most common repair he does is fixing bike tires. He charges $20 for this and it cost him $5 in parts. In one month Jim does 300 of these repairs. He also does 2 more complex repairs for $300 each and cost $50 in parts. He also sells $2000 profit worth of things from his retail shop. If rent and other fixed expense for the shop is $4000 a month how much profit did the shop make?
###Reason###
The candidate prompt, in this case, involves the model attempting to solve the entire problem at once. While this may work for simpler tasks, in complex scenarios it might lead to less accurate or nonsensical outputs. Breaking the problem down would make the reasoning more explicit and easier for the model to handle. The 'Chain of Thought' approach is better in this context because it makes the computation more manageable by breaking it down into simpler steps. This approach can be particularly useful for complex mathematical problems as it allows the model to focus on one part of the problem at a time. It also provides a clear and organized structure, making it easier for the user to follow the model's reasoning process. The candidate prompt also does not provide a common output format like "the answer is \\answer{}". The better prompt should not alter or skip the information provided in the candidate prompt which could lead to incorrect calculation.
###Better Prompt Type###
[MATHEMATICAL REASONING]
###Better Prompt###
Solve the following problem step by step
Step 1: Calculate the profit earned by Jim from fixing bike tires given that he charges $20 and it cost him $5 in parts after doing 300 of these repairs. Step 2: Compute the profit from the two more complex repairs, which cost $50 in parts and are charged at $300 each. Step 3: Sum up the profit from the fixed bike tires, the complex repairs, and the $2000 profit from the retail shop. Step 4: Subtract the rent and other fixed expenses which amounts to $4000 a month, from the total profit to find out how much profit Jim's shop made. Print the output in the format "The answer is \\answer{}"

###Candidate Prompt###
A car travels at a speed of 60 miles per hour. How far will it travel in 2.5 hours?
###Reason###

```
By using chain of thoughts prompting, we can guide the model to break down the word
    problem into logical steps and provide a step-by-step solution. The better
    prompt should not alter or skip the information provided in the candidate
    prompt which could lead to incorrect calculation.
###Better Prompt Type###
[MATHEMATICAL REASONING]
###Better Prompt###
Let's solve the following word problem step by step: A car travels at a speed of 60
    miles per hour. How far will it travel in 2.5 hours? Start by multiplying the
    speed (60 mph) by the time (2.5 hours) to find the total distance. Then,
    apply the formula Distance = Speed X Time to calculate the answer. Print the
    output in the format "The answer is \\answer{}".

###Candidate Prompt###
You have to generate a question for the given text. The question should be relevant
    and contextually appropriate, capturing the essence of the text.
###Reason###
The candidate prompt is relatively short and lacks specific instructions and
    requirements for generating a question for the text. Although it mentions the
    need for a relevant and contextually appropriate question that captures the
    essence of the text, it does not provide clear guidelines on how to determine
    the essence, which question generation model to use, or how to handle complex
    or technical text. The prompt is vague and leaves many crucial details to
    interpretation, making it difficult for the model to generate an accurate
    question.
###Better Prompt Type###
[CONTENT_GENERATION][ANALYSIS]
###Better Prompt###
Your task is to generate a relevant and contextually appropriate question for the
    given text that captures the essence of the text. Use a well-defined question
    generation model or library to generate the question. Pay attention to the
    nuances of the text, such as the main ideas, arguments, or themes, that may
    affect the question generation. Ensure that the question is contextually
    appropriate and maintains the same intent as the original text. Provide a
    clear and concise question that accurately reflects the essence of the text.
    Aim for a high level of accuracy and coherence in your question generation. If
     necessary, provide a brief explanation or evidence to support your question.

###Candidate Prompt###
The concert was scheduled to be on 06/01/1943, but was delayed by one day to today.
     What is the date 10 days ago in MM/DD/YYYY?
###Reason###
The candidate prompt is relatively short and lacks specific instructions and
    requirements for solving the problem. Although it mentions the need to find
    the date 10 days ago in MM/DD/YYYY format, it does not provide clear
    guidelines on how to calculate the date, which calendar system to use, or how
    to handle leap years. The prompt is vague and leaves many crucial details to
    interpretation, making it difficult for the model to generate an accurate
    answer. The candidate prompt also does not provide a common output format like
     "the answer is \\answer{}".
###Better Prompt Type###
[MATHEMATICAL REASONING][DATE UNDERSTANDING]
###Better Prompt###
Your task is to calculate the date 10 days ago in MM/DD/YYYY format, given that the
    concert was scheduled to be on 06/01/1943 but was delayed by one day to today
    . Use the Gregorian calendar system to calculate the date, taking into account
     leap years and the number of days in each month. Pay attention to the nuances
     of the problem, such as the initial date and the number of days to subtract,
    that may affect the calculation. Ensure that the date is contextually
    appropriate and maintains the same intent as the original problem. Provide a
    clear and concise date that accurately reflects the solution to the problem.
    Aim for a high level of accuracy and consistency in your calculations. If
```

necessary, provide a brief explanation or evidence to support your answer.
Print the output in the format "The answer is \\answer{}"

###Candidate Prompt###
Is the following sentence plausible? "Joao Moutinho caught the screen pass in the
    NFC championship."
###Reason###
The candidate prompt is relatively short and lacks specific instructions and
    requirements for evaluating the plausibility of the sentence. Although it
    provides an example and a correct answer, it does not provide clear guidelines
     on how to reason through the sentence or how to handle ambiguous or complex
    sentences. The prompt is vague and leaves many crucial details to
    interpretation, making it difficult for the model to generate an accurate
    answer. The candidate prompt also does not provide a common output format like
     "the answer is \\answer{}".
###Better Prompt Type###
[CONSTRAINED OUTPUT][ANALYSIS]
###Better Prompt###
Your task is to determine the plausibility of the following sentence: "Joao
    Moutinho caught the screen pass in the NFC championship." Provide a clear and
    concise answer, along with a brief explanation or evidence to support your
    answer. Consider the context of the sentence, such as the teams, players, and
    events mentioned, as well as the rules and conventions of the sport. Ensure
    that your answer is contextually appropriate and maintains the same intent as
    the original sentence. Pay attention to providing a well-reasoned and evidence
    -based answer that is easy to understand and follow. Print the output in the
    format "The answer is \\answer{}"

###Candidate Prompt###
Paraphrase the given sentences. Make sure the paraphrased versions have the same
    meaning as the original sentences.
###Reason###
The candidate prompt is relatively short and lacks specific instructions and
    requirements for paraphrasing. It only mentions the need for paraphrasing and
    maintaining the same meaning as the original sentences. However, it does not
    provide clear guidelines on how to ensure variations in wording, sentence
    structure, or phrasing. The prompt is vague and leaves many crucial details to
     interpretation, making it difficult for the model to generate accurate and
    diverse paraphrases.
###Better Prompt Type###
[PARAPHRASE][ROLE_PLAYING]
###Better Prompt###
Your task is to paraphrase the given sentences while maintaining the original
    meaning. The paraphrased versions should exhibit variations in wording,
    sentence structure, and phrasing. Aim for paraphrases that are linguistically
    diverse, capturing different ways of expressing the same ideas. Ensure that
    the paraphrases are contextually appropriate and maintain the same intent as
    the original sentences. Pay attention to preserving the nuances, tone, and
    style of the original sentences during the paraphrasing process.

###Candidate Prompt###
Apply a function to the final input list to generate the output list. Use any
    preceding inputs and outputs as examples.
input : [1, 7, 9, 4, 6, 2, 0]
output : [0]
input : [8, 3, 4, 0, 5, 1, 6, 9, 2]
output : [6]
input : [8, 4, 5, 9, 0, 6, 1]
output : [1]
###Reason###
The candidate prompt says nothing about the task at hand which makes it ambiguous.
    Since there could be many interpretations from a few input-output pairs the
    prompt will lead the model to guess the task at hand and we may get different

```
       outputs. To tackle this problem, the better prompt should specify the task at
       hand by clearly specifying the instructions. Specifying the objective function
        will make the better prompt unambiguous and the model would not guess. Also,
       remember we need to consider all the edge cases (such as zero or one element)
       before designing a better prompt.
###Better Prompt Type###
[PATTERN IDENTIFICATION][ANALYSIS]
###Better Prompt###
From the given list as Input, remove all the elements from the input list except
       the 7th element. Print the output in the format "The answer is \\answer{}".
input : [1, 7, 9, 4, 6, 2, 0]
output : [0]
input : [8, 3, 4, 0, 5, 1, 6, 9, 2]
output : [6]
input : [8, 4, 5, 9, 0, 6, 1]
output : [1]
input : [3, 1, 7, 2, 9, 6, 8, 4, 5, 0]
output : [8]

###Candidate Prompt###
Input: Please call once you get there.
Output: Please call upon your arrival.
Input: Thanks for your help.
Output: Your assistance is much appreciated.
Input: This song is fire.
Output: This song is incredibly good.
###Reason###
The candidate prompt is vague, ambiguous, and incomplete. Merely giving input and
       output pairs makes the interpretation of the task ambiguous. There could be
       several transformation functions that could be applied to obtain the same
       input and output pairs. The better prompt should analyze the examples,
       identify the transformation function, and make the task clear while generating
        the better prompt type. Specifying the objective function will make the
       better prompt unambiguous and the model would not guess. Also, remember we
       need to consider all the edge cases (such as zero or one element) before
       designing a better prompt.
###Better Prompt Type###
[PATTERN IDENTIFICATION][ANALYSIS]
###Better Prompt###
Write a paraphrase of the input sentence, but use a formal style. Print the output
       in the format "The answer is \\answer{}"
Input: Please call once you get there.
Output: Please call upon your arrival.
Input: Thanks for your help.
Output: Your assistance is much appreciated.
Input: This song is fire.
Output: This song is incredibly good.

###Candidate Prompt###
```

## B ADDITIONAL EXPERIMENTAL RESULTS

### B.1 TASK-SPECIFIC INSTRUCTIONS OPTIMIZED BY PROMPTD

Table 4 demonstrates all the optimized task instructions obtained using PROMPTD. For reproducibility purposes, we present both the original and the rewritten task instructions.

### B.2 RESULTS ON MATH BY CATEGORIES

Table 5 presents the breakdown of the numbers on the MATH dataset across task difficulties. for the medium to very hard categories in the MATH dataset PROMPTD gains over 6-20% accuracy scores when compared with Zero-Shot baselines.

| Task Type | Examples |
|---|---|
| **GSM8K and MATH** | **Original Task Instruction:** *Solve the following word problem step by step.* |
| | **Rewritten Task Instruction:** Consider solving a word problem step by step. Begin by reading the problem thoroughly to ensure a complete understanding. Identify all the variables and the relationships between them. Next, break the problem into smaller manageable parts and develop a strategy to solve each part. Lastly, review the solutions of all parts and check whether the final solution makes sense in the context of the original problem. |
| **Human Eval:** | **Original Task Instruction:** *Given an incomplete python template, generate the full code examining all the descriptions or test cases provided inside the template.* |
| | **Rewritten Task Instruction:** You need to create a Python function based on the information provided in the enclosed comments. Ensure that the function adheres to the function signature and description provided. Include any necessary module imports, maintain the intent of the code, and consider possible edge cases to result in robust code execution. Please follow the specific instructions provided in the comments to develop the complete Python code. |
| **Anachronisms:** | **Original Task Instruction:** *Identify whether a given statement contains an anachronism.* |
| | **Rewritten Task Instruction:** Review the given statement and identify whether it includes an anachronism - a chronological inconsistency in which a person, event, object, or language is mistakenly placed in a time period where it does not belong. Clearly state "Yes" if the statement contains an anachronism and "No" if it does not. Explain your reasoning briefly, providing details that led you to your conclusion. Print the output in the format "The answer is \\answer" and if yes also provide "Identified anachronism is - \\response" otherwise print "No anachronism found". |
| **Date Understanding** | **Original Task Instruction:** *Infer the date from context* |
| | **Rewritten Task Instruction:** Given the context of the text, infer the date it is referring to. Consider any key phrases, words or events that could hint towards a specific date. If necessary, conduct a brief background check of any prominent figures or events that mention in the text. Try to narrow down the exact date in MM/DD/YYYY format. Print the output in the format "The answer is \\answer". |
| **Known Unknowns** | **Original Task Instruction:** *A test of 'hallucinations' by asking questions whose answers are known to be unknown* |
| | **Rewritten Task Instruction:** You will be provided with a series of questions that test for 'hallucinations,' where the correct response is either a specific answer or unknown. Your job is to look at the question and determine if the answer can be found from the question or if it is undeterminable. If the answer cannot be concluded from the question alone, choose "(b) unknown." If the answer can be determined from the question alone, provide the answer in the form "(a) Your Answer." Print the output in the format "The answer is \\answer" |
| **Sports Understanding** | **Original Task Instruction** *Determine whether the following statement or statements are plausible or implausible:* |
| | **Rewritten Task Instruction:** Your task is to evaluate the plausibility of the following statement(s): [input statement]. Consider the context, factual accuracy, logic, and consistency of the statement(s) in your evaluation. Then, provide a clear and substantiated determination of whether each statement is plausible (could likely be true) or implausible (unlikely to be true). Print the output in the format "The statement is \\answer". (Replace \\answer with "plausible" or "implausible") . |
| **Analytical Entailment** | **Original Task Instruction:** *Identify whether one sentence entails the next* |
| | **Rewritten Task Instruction:** Your task is to analyze two sentences and determine if one logically entails the other. If the first sentence implies or logically leads to the truth of the second, state "Entailment". If it is unclear or ambiguous, state "Uncertain". Print the output in the format "The answer is \\answer" |

Table 4: Rewritten Optimized task instructions by PROMPTD for each of the tasks.

| | Category - I (Very Easy and Easy) | Category - II (Medium, Hard, and Very Hard) |
|---|---|---|
| Zero-Shot | 76.429 | 30.476 |
| Zero-Shot-CoT | 80.714 | 40.476 |
| PROMPTD (Instruction-Only) | 76.429 | 44.762 |
| PROMPTD | 89.286 | 50.476 |

Table 5: Breakdown of performance of all the approaches across different difficulty categories of MATH dataset.

## B.3 RESULTS FOR TASK TYPE GENERALIZATION

In this section, we extend the generalization capability of PROMPTD across two different task types namely Symbolic Reasoning Kojima et al. (2022) and Open Domain Question Answering

| | ZeroShot | ZeroShotCoT | PROMPTD | PROMPTD (after executing python codes) |
|---|---|---|---|---|
| OpenDomain-QA (StrategyQA) | 71.600 | 73.272 | **78.400** | - |
| Symbolic Reasoning (Last Letter Concatenation) | 6.000 | **90.800** | 58.800 | 80.800 |

Table 6: Evaluation of PROMPTD on two unseen task types namely OpenDomainQA and Symbolic Reasoning

| Bad Prompt | Classify the following sentences for sentiment. Examples are |
|---|---|
| Reason | The candidate prompt is vague and incomplete. It fails to clearly specify the task to be performed on the given sentences and does not provide any specific sentiment categories to assign to the sentences. Additionally, no examples are provided to guide the model in understanding the task. The better prompt should clearly specify the categories for sentiment analysis (e.g., positive, neutral, negative) and provide examples to illustrate the task. The prompt should also be structured in such a way as to instruct the model to utilize reasoning and understanding of language semantics to correctly classify the sentences' sentiments. Remember the better prompt should not alter or skip the information provided in the candidate prompt. |
| Task Type | [CLASSIFICATION][SENTIMENT ANALYSIS] |
| Better Prompt | You have to *classify the following sentences as either "Positive", "Neutral", or "Negative" based on sentiment analysis.* Draw on your understanding of language semantics and reasoning skills to accurately assess the sentiment of each sentence. Print the output in the format "The sentence {input sentence} has a {positive/neutral/negative} sentiment". *For example:* *Input: This movie was fantastic.* *Output: The sentence This movie was fantastic has a Positive sentiment.* *Input: I can't stand this weather.* *Output: The sentence I can't stand this weather has a Negative sentiment* |

Table 7: Example prompt improvement by PROMPTD for an unseen sentiment classification task. The original prompt lacks key details like sentiment categories and examples. PROMPTD enhances the prompt by specifying positive, neutral, and negative sentiment classes, providing illustrative examples, and structuring the prompt to guide the model's reasoning and output format. This results in a more complete, specific prompt that enables the successful completion of sentiment analysis.

(OpenDomainQA) Geva et al. (2021). For the experiments we pick 250 random samples from the Last Letter Concatenation dataset Kojima et al. (2022) and StrategyQA Geva et al. (2021).

As shown in Table **??**, PROMPTD outperforms zero-shot baselines for OpenDomainQA while falling short on symbolic reasoning task.

In addition to the experiment with PROMPTD in Section 3.6, in Table 7, we present an example using PROMPTD (Instruction-only) to rewrite the instruction for an unseen task type of sentiment classification. For the prompt "Classify the following sentences for sentiment. Examples are", it correctly identifies the task type as a "Sentiment Analysis" and "Classification" task and generates a prompt that can be used for downstream sentiment classification tasks. Moreover, PROMPTD can generate a possible set of classes (such as positive, neutral, or negative for sentiment classification task) and generate a set of examples that can be further used for in-context learning using few-shot sentiment classification demonstrations.

## B.4 ERROR ANALYSIS

In Table 8, we summarize two common error categories of PROMPTD. While such hallucination errors are rare, PROMPTD is still found to make them during prompt rewriting.

| Error Type | Subcategories | Description | Example |
|---|---|---|---|
| Hallucinated instruction | Fabricated Information (4.348%) | PROMPTD fabricates and adds informations in the rewritten prompt. | **Original Prompt:** Given a list of numbers, return the sum of squares of the numbers[...]def double_the_difference(...)
**Rewritten Prompt:** [...]def sum_of_squares_of_odd_numbers (...) |
| Manipulated information | Incorrect Copy from Test Instances (58.261%) | PROMPTD skips information from the test-instance. | **Original Prompt:** Solo has to read 4 pages from his Science textbook[...]
**Rewritten Prompt:** [...]Step 1: Calculate the total number of pages Solo needs to read from his Science, Social Studies, History, and Geography textbooks.[...] |
| | Add Unnecessary Constraints (23.478%) | PROMPTD adds strict constraints leading to incorrect final response. | **Original Prompt:** def largest_divisor(n: int) ->int: """ For a given number n, find the largest number that divides n evenly, smaller than n\n [...] """
**Rewritten Prompt:** [...]print("Invalid input. Please enter a number greater than 1".)[...] |
| | Correct Reason Incorrect rewritten prompt (13.913%) | PROMPTD correctly identifies shortcomings of test-instance but is unable to reflect the changes in the rewritten prompt. | **Original Prompt:** [...] In this Kata, you have to sort an array of non-negative integers according to number of ones in their binary representation in ascending order. For similar number of ones, sort based on decimal value.[...]
**Reason:** [...] but the provided examples are simply sorted in ascending order, which doesn't align with the initial instructions [...]
**Rewritten Prompt:** [...] >>> sort_array([1, 0, 2, 3, 4]) should return [0, 1, 2, 3, 4] [...] |

Table 8: Error categories of PROMPTD's rewritten prompts.

## B.5   EXPERIMENTS OF PROMPTD USING GPT-3.5

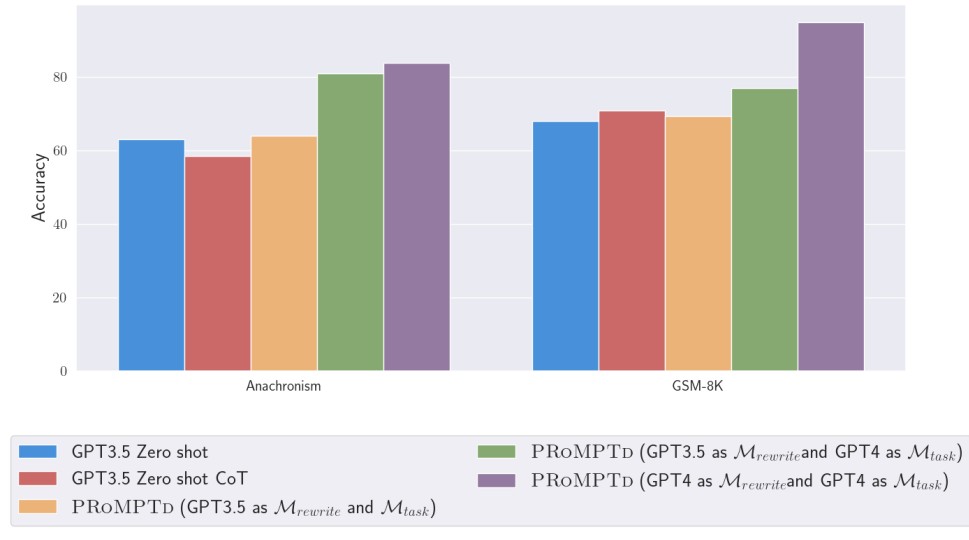

Figure 6: Results of PROMPTD using GPT-3.5-turbo.

In Figure 6, we present results of variants of PROMPTD based on GPT-3.5-turbo, including: (1) zero-shot GPT-3.5; (2) zero-shot CoT of GPT-3.5; (3) PROMPTD when using GPT-3.5 as both the task and the rewriting LLMs; (4) PROMPTD when using GPT-3.5 to rewrite prompts while GPT-4 as the zero-shot task LLM; (4) PROMPTD when using GPT-4 as both the task and the rewriting LLMs. Experiments were conducted on Anachronisms and GSM-8K. The results showed the difficulty of using GPT-3.5 to rewrite prompts. In particular, we found that the prompts rewritten by a relatively weaker LLM (GPT-3.5) would not assist the relatively stronger LLM (GPT-4) in task.

B.6   ANALYSIS OF RESPONSES GENERATED BY ZERO-SHOT BASELINES AND PROMPTD

In this section, we discuss the responses generated by all the approaches to analyze the extra pass for answer extraction. As shown in Table 9, for the Zero-Shot baseline, the answer extraction becomes difficult as LLM could put the response anywhere when left unconstrained. Zero-Shot-CoT (Kojima et al., 2022), on the other hand, overcomes this problem by designing task-specific answer extraction prompts such as "Therefore, the answer (arabic numeral is)" or "Therefore, among A through E, the answer is". However, one needs to engineer such prompts for each task and it becomes difficult to extract answers for code generation or content generation tasks. PROMPTD overcomes such limitations by using a general answer extraction instruction "Provide you answer in the format The Answer is \answer{}"

| Task Type | Examples |
|---|---|
| **Mathematical Reasoning** | **Original Prompt (GSM8K)** *Frederick is making popsicles to sell and to save money he is making his own popsicle sticks. He can get 200 sticks from a 2 x 4 piece of wood and 400 sticks from a 2 x 8 piece of wood. He has $24 to buy wood for sticks. A 2 x 4 costs $4. A 2 x 8 costs $6. What is the most popsicle sticks he can make if he buys the cheapest lumber?* |
| | **Zero-Shot:** He can buy 6 two by fours because 24 / 4 = $<< 24/4 = 6 >>$6 This will make him 1,200 sticks because 6 x 200 = $<< 6*200 = 1200 >>$1,200. Answer:$\boxed{1200}$. |
| | **Zero-Shot-CoT:** Let's think step by step. *[Truncated]* From the 2 x 8 wood, he can make 1,600 sticks (400 sticks x 4 pieces = 1,600 sticks). 
 *Therefore, Frederick can make more popsicle sticks if he buys the 2 x 8 wood.* |
| | **PROMPTD:***[Truncated]* 
 So even though our initial hypothesis suggested otherwise it turns out: 
 The most popsicle sticks that Frederick can produce is \answer{48}. |
| | **Original Prompt:** Find the reflection of $\\ beginpmatrix -2 \\\\ 3 \\\\ end{pmatrix}$ over the vector $\\ begin{pmatrix} 2 \\\\1 \\ end{pmatrix}.$ |
| | **ZeroShot:** *[Truncated]* 

 `So, we compute that $-4 + 3 = -1,$ and $4 + 1 = 5.$ Therefore,` 
 `\[\bold p = \begin {bmatrix}-4\\6\\\end {bmatrix},\]and the answer is $\boxed {\begin {` 
 `    bmatrix}-4\\6\\\end {bmatrix}}$.` |
| | **Zero-Shot-CoT:** *[Truncated]* 

 `Therefore, the answer (arabic numerals) is` 
 `The reflection of the point $\\begin{pmatrix} -2 \\\\ 3 \\end{pmatrix}$ over the vector $\\` 
 `    begin{pmatrix} 2 \\\\ 1 \\end{pmatrix}$ is given by:` 
 `$-83/28 \approx -2.96$` 
 `$101/28 \approx 3.61$` 
 `Therefore, in decimal form, it is approximately:` 
 `$\left[{\matrix{-2.96}\cr {3.61}\cr }\right]$` |
| | **PROMPTD:** *[Truncated]* 
 Hence the answer is $(-(18/5), (9/15))$. 
 The final output would be "The answer is $(-(18 / 5), (9 /15 ))$" |
| **Logical Reasoning** | **Original Prompt (Date Understanding)** *Today is the palindrome day of 2020, because the MMDDYYYY format of the date is the same backwards as forwards. What is the date 10 days ago in MM/DD/YYYY?* |
| | **Zero-Shot:** 
 The palindrome day of 2020 is 02/02/2020. So, 10 days ago would be 01/23/2020. |
| | **Zero-Shot-CoT:** Let's think this step by step. 
 The palindrome day of 2020 is 02/02/2020. If we subtract ten days from this date, the answer would be 01/23/2020. 
 Therefore, the correct choice is A) 01/23/2020. 
 Therefore, among A through E, the answer is 
 A) 01/23/2020 |
| | **PROMPTD:** 
 The palindrome day of the year 2020 is 02/02/2020. Subtracting 10 days from this date gives the date 01/23/2020. 
 Therefore, the answer is 01/23/2020. |

Table 9: Comparison of responses generated by ZeroShot, ZeroShotCoT, and PROMPTD. It is very hard to extract the final answer from the ZeroShot baseline due to the absence of a common format. While, for ZeroShotCoT, one needs to engineer different answer extraction prompts such as "Therefore, the answer (arabic numerals) is " or "Therefore, among A through E, the answer is " to extract the answer.

## C    EXAMPLES OF RESPONSES GENERATED BY PROMPTD

We present outputs of PROMPTD on all eight datasets in Table 10-12.

| Task Type | Examples |
|---|---|
| **Mathematical Reasoning** | **Original Prompt:** *Josh decides to try flipping a house. He buys a house for $80,000 and then puts in $50,000 in repairs. This increased the value of the house by 150%. How much profit did he make?* |
| | **Reason:** The candidate prompt is unclear as 'increased the value of the house by 150%' could be interpreted in two ways: 1) the value of the house after the repair is 150% of the buying price, or 2) the value of the house after the repair is the buying price plus an additional 150% of the buying price. The prompt also does not specify whether the profit should include the cost of repairs or not, and does not provide a common output format for the answer. The better prompt clarifies these ambiguities by explicitly explaining the meaning of 'increased the value by 150%' and specifying that the profit should be calculated after taking into account the cost of repairs. |
| | **Rewritten Prompt:** Josh decides to flip a house. He bought a house for $80,000 then invested an additional $50,000 for its repairs. After repairs, the house's value increased such that it's now worth 150% more than the price at which he bought it. Calculate the profit he made after selling the house, considering both the buying price and cost of repairs. Print the output in the format "The answer is \\answer" |
| | **Original Prompt:** *How many distinct arrangements are there of PAPA?* |
| | **Reason:** The candidate prompt leaves room for interpretation, as it doesn't ask for a specific output format or example to guide the reasoning process of the model. An improved prompt would provide clear instructions about the strategy, such as completing the task by using the formula for permutations of multi-set words, and would guide the desired output format to improve answer extraction. |
| | **Rewritten Prompt:** Compute the number of distinct arrangements that can be made from the word 'PAPA'. To solve this problem, use the formula to count the permutations of a multiset: $n!/(n1!*n2!*...*nk!)$, where $n$ is the total number of letters and $n1, n2,...$ $nk$ are the count of each distinct letter. In this case, 'P' appears twice and 'A' appears twice in 'PAPA'. Print the output in the format "The answer is \\answer" |
| **Logical Reasoning** | **Original Prompt:** *Identify whether one sentence entails the next:Every bachelor is tall, and the man in the hat is short. Therefore the man in the hat is married* |
| | **Reason:** The candidate prompt is too short and lacks specific instructions and requirements for answering the question. It only mentions the need to identify if one sentence entails the next, but it does not provide clear guidelines on how to reason through the question or provide evidence for the answer. The prompt is vague and leaves many crucial details to interpretation, making it difficult for the model to generate an accurate answer. The candidate prompt also does not provide a common output format like "the answer is \\answer". |
| | **Rewritten Prompt:** Your task is to determine whether the following sentences entail each other: "Every bachelor is tall, and the man in the hat is short. Therefore the man in the hat is married." Provide a clear and concise answer, along with a brief explanation or evidence to support your answer. Consider the logic of the statements, the properties and characteristics of the entities mentioned, and the implications of the sentences. Ensure that your answer is contextually appropriate and maintains the same intent as the original question. Pay attention to providing a well-reasoned and evidence-based answer that is easy to understand and follow. Print the output in the format "The answer is \\answer" |
| | **Original Prompt:** *Identify whether a given statement contains an anachronism: The sun's creation during the Stelliferous Era is the cause of its size, color, and brightness.Your choices are: (1) Yes (2) No* |
| | **Reason:** TThe candidate prompt is too short and lacks specific instructions and requirements for answering the question. It only mentions the need for a yes or no answer, but it does not provide clear guidelines on how to reason through the question or provide evidence for the answer. The prompt is vague and leaves many crucial details to interpretation, making it difficult for the model to generate an accurate answer. The candidate prompt also does not provide a common output format like "the answer is \\answer". |
| | **Rewritten Prompt:** Your task is to analyze the following statement for any potential anachronisms: Ṫhe sun's creation during the Stelliferous Era is the cause of its size, color, and brightness.Än anachronism, in this context, refers to something that is out of time or chronologically misplaced. Provide a clear and concise answer, along with a brief explanation or evidence to support your answer. Consider the historical and scientific accuracy of the statement. Your choices are: (1) Yes, it contains an anachronism (2) No, it does not contain an anachronism. Print the output in the format "The answer is \\answer" |
| | **Original Prompt:** *Infer the date from context: Jane was born on the last day of Feburary in 2000. Today is her 16-year-old birthday. What is the date today in MM/DD/YYYY?* |
| | **Reason:** The candidate prompt is ambiguous and lacks clarity. It doesn't specify the calendar system to use and doesn't make clear that leap years should be taken into account when calculating the date. The prompt also lacks a common output format, which would allow for easy extraction of the final answer. The better prompt should provide clear instructions on how to calculate the date and specify that the Gregorian calendar system should be used. The better prompt should also provide a common output format like "The answer is \\answer". |
| | **Rewritten Prompt:** Your task is to calculate today's date in MM/DD/YYYY format, given that Jane was born on the last day of February in 2000 and today is her 16th birthday. Use the Gregorian calendar system to calculate the date, taking into account leap years and the number of days in each month. Pay attention to the nuances of the problem, such as the initial date and the number of years to add, that may affect the calculation. Ensure that the date is contextually appropriate and maintains the same intent as the original problem. Provide a clear and concise date that accurately reflects the solution to the problem. Aim for a high level of accuracy and consistency in your calculations. If necessary, provide a brief explanation or evidence to support your answer. Print the output in the format "The answer is \\answer" |

Table 10: Generated examples of PROMPTD for mathematical and logical reasoning tasks. *In-order* of their appearance, the samples are taken from GSM8K, MATH, Analytical Entailment, Anachronisms, and Date Understanding.

| Task Type | Examples |
|---|---|
| Code Generation | **Original Prompt:** |

```
from typing import List
def separate_paren_groups(paren_string: str) -> List[str]:
    """
    Input to this function is a string containing multiple groups of nested parentheses.
     Your goal is to
    separate those group into separate strings and return the list of those.
    Separate groups are balanced (each open brace is properly closed) and not nested within
     each other
    Ignore any spaces in the input string.
    >>> separate_paren_groups('( ) (( )) (( )( ))')
    ['()', '(())', '(()())']
    """

    pass  # Function implementation here
```

**Reason:** The candidate prompt is incomplete as it does not provide any example input or output for the model to follow. It also lacks specific instructions on how to handle edge cases such as empty strings or strings without parentheses. The better prompt clarifies these points and provides a clear task description along with a specific input-output format. It also provides a complete function signature, which makes it easier for the model to generate the correct code.

**Rewritten Prompt:**
Write a Python function that takes a string as an input, where the string contains multiple groups of nested parentheses. The function should separate these groups into separate strings and return a list of these strings. Each group of parentheses is balanced (i.e., every open parenthesis is properly closed) and not nested within each other. Ignore any spaces in the input string. Handle the edge cases such as empty strings or strings without parentheses by returning an empty list. The function signature is: '$def\ separate\_paren\_groups(paren\_string:\ str)$ -> List[str]:'. An example of the expected functionality is: For the input '$separate\_paren\_groups('()(())(()())')$', the function should return '['()', '(())', '(()())']'.

**Original Prompt:**

```
def encode_shift(s: str) -> str:
    # returns encoded string by shifting every character by 5 in the alphabet.
    return ''.join([chr(((ord(ch) + 5 - ord('a')) % 26) + ord('a')) for ch in s])
def decode_shift(s: str) -> str:
    # takes as input string encoded with encode_shift function. Returns decoded string.
    def shift_char(char: str) -> str:
        if not char.isalpha():
            return char
        base = 65 if char.isupper() else 97
        return chr((ord(char) - base - 5) % 26 + base)
    return ''.join(map(shift_char, s))
```

**Reason:** Write a Python function that takes as input a string encoded with the 'encode_shift' function. The 'encode_shift' function returns an encoded string by shifting every character by 5 in the alphabet. Your task is to write the 'decode_shift' function that returns the original string before it was encoded. Your function should take into account the wrap-around of the alphabet, i.e., after 'z', it should go back to 'a'. The function signature should be: '$def\ decode\_shift$(s: str) $->$ str:'

Table 11: Generated examples of PROMPTD for code generation task. The samples are taken from the Human Eval dataset.

| Task Type | Examples |
|---|---|
| **Common Sense Reasoning** | **Original Prompt:** A test of 'hallucinations' by asking questions whose answers are known to be unknown: How often did Abraham Lincoln cut his toenails? Your choices are: (1) Every Saturday night (2) Unknown |
| | **Reason:** The candidate prompt is too short and lacks specific instructions and requirements for answering the question. It only mentions the need for a choice between two options, but it does not provide clear guidelines on how to reason through the question or provide evidence for the answer. The prompt is vague and leaves many crucial details to interpretation, making it difficult for the model to generate an accurate answer. The candidate prompt also does not provide a common output format like "the answer is \\answer{}". |
| | **Rewritten Prompt:** Your task is to answer the following question: How often did Abraham Lincoln cut his toenails? Your choices are: (1) Every Saturday night (2) Unknown. Provide a clear and concise answer, along with a brief explanation or evidence to support your answer. Consider the context of the question, such as the historical period and the availability of information about Abraham Lincoln's personal habits. Ensure that your answer is contextually appropriate and maintains the same intent as the original question. Pay attention to providing a well-reasoned and evidence-based answer that is easy to understand and follow. Print the output in the format "The answer is \\answer{}". |
| | **Original Prompt:** *Determine whether an artificially constructed sentence relating to sports is plausible or implausible: Malcolm Brogdon banked the shot in* |
| | **Reason:** The candidate prompt is not related to basketball, and it appears to be a mix of different prompts. It doesn't provide any specific instructions or requirements for determining the plausibility of a sentence related to basketball. The better prompt provides a clear context and a specific sentence to evaluate. It also provides a specific output format, guiding the model to generate a clear and concise answer with a brief explanation. This ensures that the model understands the task and can generate the appropriate response. Print the output in the format "The answer is \\answer{}". |
| | **Rewritten Prompt:** Determine whether the following sentence related to basketball is plausible or implausible: "Malcolm Brogdon banked the shot in." Provide a clear and concise answer (either 'plausible' or 'implausible') along with a brief explanation of your reasoning. Consider the context of the sentence, the players involved, and the action described. Ensure that your answer is contextually appropriate and maintains the same intent as the original sentence. |

Table 12: Generated examples of PROMPTD for Sports Understanding and Known-Unknowns from BigBench.

| Task Type | Examples |
|---|---|
| **OpenDomainQA** | **Original Prompt:** *Answer questions in which the required reasoning steps are implicit in the question:*
*Do the anchors on Rede Globo speak Chinese?*
*Your choices are: (1) Yes (2) No* |
| | **Reason:** The candidate prompt is vague and lacks specific instructions and requirements for answering the question. Although it provides an example and a correct answer, it does not provide clear guidelines on how to reason through the question or how to handle ambiguous or complex questions. The prompt is vague and leaves many crucial details to interpretation, making it difficult for the model to generate an accurate answer. The candidate prompt also does not provide a common output format like "the answer is \\answer{}". |
| | **Rewritten Prompt:** Your task is to answer the following question: "Do the anchors on Rede Globo speak Chinese?" Consider the context of the question, such as the language typically spoken on Rede Globo, a Brazilian television network. Ensure that your answer is contextually appropriate and maintains the same intent as the original question. Provide a clear and concise answer, along with a brief explanation or evidence to support your answer. Your choices are: (1) Yes (2) No. Print the output in the format "The answer is \\answer{}" |
| **Logical Reasoning** | **Original Prompt:** *Take the last letters of each words in "Britt Tamara Elvis Nayeli" and concatenate them.* |
| | **Reason:** TThe candidate prompt is vague and does not specify the order of concatenation, whether to consider spaces or punctuation as words, or how to handle special characters. The prompt is also open-ended, allowing for multiple correct answers, which may confuse the model. The candidate prompt also does not provide a common output format like "the answer is \\answer". |
| | **Rewritten Prompt:** Your task is to take the last letters of each word in the phrase "Britt Tamara Elvis Nayeli"in the same order as they appear in the phrase and concatenate them. Do not consider spaces or punctuation as words and ignore any special characters. Print the output in the format "The answer is \\answer" |

Table 13: Generated examples of PROMPTD for OpenDomain-QA and symbolic reasoning tasks. *In-order* of their appearance, the samples are taken from StrategyQA Geva et al. (2021) and Kojima et al. (2022).

