# OpenReview forum: "Instance Needs More Care: Rewriting Prompts for Instances Yields Better Zero-Shot Performance"
_ICLR.cc/2024/Conference — Submitted to ICLR 2024_

### Official Review · Reviewer_bonY · 2023-11-01

**Soundness:** 2 fair
**Presentation:** 2 fair
**Contribution:** 2 fair
**Rating:** 3
**Confidence:** 3

**Summary:**

This paper studies the quailty of prompts and how to improve them for better capabilities of zero-shot and few-shot in-context learning. To ahieve this, authors first define the characteristics that good prompts should have, and then proposes a method to rewrite prompts to improve their quality based on this. Effectiveness is evaluated on mathematical reasoning, code generations and those tasks in BigBench.

**Strengths:**

I think the tageted issue, i.e., the quality and improvements of prompts, is currently needed by both the industrial and academic communities. Improving the zero-shot or few-shot in-context learning ability of large-scale models by improving the quality of prompts still has extremely high research value in the short term. This work could be viewed as a start point for this aspect.

However, we have to realize that the research space in prompt engineering also reflects the shortcomings of the current large models. The improvement of large models in the near future will be reflected in their stronger robustness to prompts. I suggest researchers to look at this issue with a more long-term developmental perspective, instead of being satisfied with the immediate results.

**Weaknesses:**

I very much agree with the starting point of this article, but at the same time, I regret that this research work lacks the necessary depth. From the definition of the quality of prompts to the method of improving the quality of prompts, most of the content is confined to quantitative analysis, lacking more in-depth and specific method design. There is also no larger scale quantitative evaluation on more general downstream tasks. For readers, it is somewhat difficult to catch the technical insights and contribution so that the current version seems premature.

**Questions:**

Could you further clarify how you rewrite the prompts and clearly hightlight the insights in them?

---

> ### Author Response · Authors · 2023-11-20
> **Response to Reviewer bonY**
>
> We appreciate the reviewer for acknowledging the value of our work and suggesting the long-term developmental perspective about LLMs’ robustness to prompts! We agree with the reviewer and found it an important direction to pursue.
>
>
> **Regarding the reviewer’s suggestion on improving LLMs’ robustness to prompts as future direction**
>
> We agree with the reviewer on the importance of improving LLMs’ robustness to prompts. In fact, this is also our motivation for this work. In our submission, we contribute to this aim by rewriting prompts that can be executed across a diverse set of tasks on different models. That is, instead of improving the robustness nature of an individual LLM, we approach this “robustness to prompts” by bridging the gap between human-written prompts (which could be suboptimal) and the LLM-preferable prompts through PROMPTD’s optimization. We did three experiments on our proposed method’s capability on the three points: robustness to unseen tasks, adaptability on weaker LLMs, and sustainability to adversarial prompting through the interpretability of PROMPTD.
>
>
> We show that PROMPTD could rewrite test instances with specialized instructions and hints across unseen task types such as implicit reasoning, symbolic reasoning, and sentiment classification. For adaptability, we show the efficacy of our approach with evolving LLMs, including GPT-3.5-turbo, and LLAMA2 70B. Finally, we present a use-case of PROMPTD on its applicability to avoid adversarial prompting because of its interpretable approach to rewriting. PROMPTD not only aligns with the current trajectory of LLM development but also actively contributes to enhancing their robustness to a diverse set of prompts aligning with the future direction suggested by the reviewer.
>
>
> **Clarification on how to rewrite prompts and their insights**
>
> Thank you for seeking further clarification on PROMPTD's methodology and insights. In summary, PROMPTD improves the zero-shot performance of LLMs by tailoring prompts to each test instance based on criteria such as specificity and structure, as demonstrated through our extensive experiments. This approach not only enhances clarity and interpretability but also showcases PROMPTD's adaptability across diverse task types.
>
>
> **Rewriting Criteria:** PROMPTD employs specificity, non-ambiguity, completeness, and structuredness as its rewriting criteria, ensuring each prompt is optimally customized for its corresponding test instance.
>
> **In-Context Learning for Rewriting:** Using a few-shot in-context learning method, PROMPTD contrasts good and bad prompts to guide the LLM in generating better prompts.
>
> **Instance-Level Customization:** Each test instance is individually considered, allowing PROMPTD to address unique requirements effectively, contrasting with generic, one-size-fits-all prompts.
>
>
> We picked the ten rewriting demonstrations for PROMPTD based on the most frequently researched task types such as Mathematical Reasoning, Logical Reasoning, Code Generation, and Instruction Induction to probe LLMs. Since we aim to improve on broad task types such as general queries to chatGPT, we also included content generation prompts. The intuition behind including such diverse tasks in demonstrations was to cover the diverse shortcomings of different task types and generalize to unseen task types. Moreover, we hypothesize that different task types could learn from different errors and when reasoning over a test instance they can learn from both the similar types of tasks and general content generation tasks. Guiding through in-context learning examples and harnessing the text generation capabilities of LLMs enables us to not only identify the limitations but also add instance-specific hints and examples in the rewritten prompts.
> We also pick several examples from the Instruction Induction task that would encourage LLMs to learn from input-output pairs and generate examples. As shown in Table 5, for the sentiment classification task, PROMPTD was able to generate several examples to learn the task. Finally, for some task types such as Code Generation and Content Generation, we refrain from generating a structured output such as “The answer is …” enabling LLMs to learn when a structured output is required.
> Each of these prompt types addresses a unique set of challenges in prompt rewriting. Together, they provide a comprehensive training ground for the model to learn a wide array of skills necessary for generating high-quality, instance-specific prompts. These diverse examples enable the model to generalize these skills across tasks, leading to better overall performance and adaptability to a range of unseen task types.

---

> > ### Comment · Reviewer_bonY · 2023-11-23
> > **Response to Authors**
> >
> > Thanks for the response. After reading the reviews and responses, I tend to keep my rating.

---

### Official Review · Reviewer_xGQ6 · 2023-11-03

**Soundness:** 3 good
**Presentation:** 3 good
**Contribution:** 3 good
**Rating:** 5
**Confidence:** 3

**Summary:**

This paper addresses a significant challenge in the application of large language models (LLMs) to zero-shot tasks – the design of task prompts that are sufficiently informative and unambiguous to guide the model to the correct solution without task-specific annotations. The authors propose PROMPTD, an innovative approach that generates customized prompts for each test instance by designing some pre-defined prompts, enhancing the LLM's ability to handle tasks across various domains including arithmetics, logical reasoning, and code generation.

**Strengths:**

1. The idea of dynamically rewriting prompts for individual instances is a novel and interesting approach that represents a significant departure from the more static strategies employed in prior works.

2. The reported results indicate that PROMPTD provides a substantial boost in performance, achieving an improvement of 10% on the MATH dataset and 5% on the code generation tasks in HumanEval, which is impressive and suggests that the approach has practical value.

3. By applying PROMPTD across eight different datasets, the authors demonstrate the method's general applicability, an essential characteristic for real-world deployments.

4. The paper presents an additional benefit of using PROMPTD – the rewritten prompts not only aid in task resolution but also enhance the interpretability of the LLM's decision-making process, which could be crucial for trust and reliability in AI systems.

**Weaknesses:**

1. The paper does not sufficiently discuss the computational overhead of the PROMPTD method. Since the approach involves generating custom prompts for each instance, there may be a significant increase in the computational cost that could limit its scalability. More importantly, the PROMPTD is quite long; can the authors make some ablation studies about it?

2: The efficacy of PROMPTD is likely highly dependent on the initial quality of the prompts it is based upon. The paper could better address how the system performs with suboptimal base prompts and the robustness of the method to variations in prompt quality.

3: While the performance improvements are impressive, the evaluation might benefit from a deeper analysis of where and why the approach fails. Understanding the limitations of PROMPTD is as important as understanding its strengths.

4: There have been so many zero-shot prompting methods recently. The paper would be strengthened by including a more comprehensive comparison with the recent state-of-the-art methods for zero-shot learning.


5. The evaluation of PROMPTD on a single new task type (sentiment classification) is a significant limitation. Given the length and complexity of the original PromptD, it is unclear how well the method would generalize to other task types. The reviewer's expectation of a more general prompt applicable to a wide array of task types is unmet. This is a critical aspect, as the creation of highly specialized prompts may not be feasible in many real-world applications.

**Questions:**

Same as before

---

> ### Author Response · Authors · 2023-11-20
> **Response to Reviewer xGQ6**
>
> We appreciate the reviewer’s acknowledgment of PROMPTD on its novelty to dynamically rewrite prompts and recognition of its interpretability to perform prompt optimization for zero-shot approaches.
>
> **W1: Computational overhead of PromptD**
>
> We refer the reviewer to our general response to this comment.
>
>
> **W2: Reliance on Prompt Quality and Robustness**
>
> We clarify that PROMPTD does not rely on the initial quality of the prompts it is based upon. We note that, as we focus on zero-shot prompting, the “base prompt” in our experiments is simply the task instruction from the original test benchmarks (see examples in Table 2), and we make no assumption on its quality.
>
>
> Our approach PROMPTD does not assume optimal “good prompts” in prompt rewriting either. Instead, PROMPTD relies on its ability to reason over the shortcomings of a given “bad prompt” and overcome them in rewriting. This ensures robust performance even with suboptimal starting prompts, as demonstrated across various tasks and datasets, including unseen ones.
>
>
>
>
> **W3: A deeper analysis of where and why the approach fails**
>
> We refer the reviewer to the general discussion on in-depth analysis.
>
> **W4: Comparison with other zero-shot approaches**
>
> Thank you for your comment regarding the comparison with other zero-shot approaches. Indeed, the approaches for zero-shot learning in LLMs are rich and rapidly expanding. To address your point, we have identified several recent papers that present innovative zero-shot techniques. Wang et al., 2023 introduced a two-step approach that plans out subtasks and then executes the plans to derive the final answer. On the contrary, our work focuses on enriching prompts with hints and encouraging LLMs to generate rationales before deriving the final answer. Reynolds and McDonell2021 suggested a meta-prompt “Let's solve this problem by splitting it into steps.” to facilitate multi-step reasoning in simple arithmetic problems. However, they did not evaluate quantitatively on diverse reasoning tasks against baselines. In this paper, we evaluate PROMTPD on a diverse set of tasks beyond reasoning. Wei et al., 2021 proposed instruction-tuning which gives models an instruction describing a task while with PROMPTD, on the other hand, instead of tuning an instruction we aim to optimize task-specific instructions and test instances. Recent approaches, similar to PROMPTD,  have also proposed to optimize task-level instructions (Hong et al., 2023) enriching task-specific hints. Through experiments, we show that such task-specific instruction optimization does not lead to stable performance across different tasks motivating the need for instance-level prompt optimization. Huang et al., 2022 show that reasoning performance improves by multi-task finetuning a model on several self-generated synthetic CoT datasets. We, on the other hand, do not perform finetuning and in addition to mathematical and logical reasoning, also focus on content generation and code generation tasks.
>
>
> **References:**
>
> Lei Wang, Wanyu Xu, Yihuai Lan, Zhiqiang Hu, Yunshi Lan, Roy Ka-Wei Lee, Ee-Peng Lim "Plan-and-solve prompting: Improving zero-shot chain-of-thought reasoning by large language models." In Proceedings of the 61st Annual Meeting of the Association for Computational Linguistics. Association for Computational Linguistics.
>
> Jiaxin Huang, Shixiang Shane Gu, Le Hou, Yuexin Wu, Xuezhi Wang, Hongkun Yu, & Jiawei Han. (2023). Large Language Models Can Self-improve. International Conference on Learning Representations. 2023.
>
> Sun, Hong, et al. "AutoHint: Automatic Prompt Optimization with Hint Generation." Workshop on Foundations and Applications in Large-scale AI Models -Pre-training, Fine-tuning, and Prompt-based Learning, colocated with the 29TH ACM SIGKDD CONFERENCE ON KNOWLEDGE DISCOVERY AND DATA MINING (KDD), August 6-10, 2023, Long Beach, CA, USA
>
> Wei, Jason, et al. "Finetuned Language Models are Zero-Shot Learners." International Conference on Learning Representations. 2021.
>
> Laria Reynolds and Kyle McDonell. 2021. Prompt Programming for Large Language Models: Beyond the Few-Shot Paradigm. In Extended Abstracts of the 2021 CHI Conference on Human Factors in Computing Systems (CHI EA '21). Association for Computing Machinery, New York, NY, USA, Article 314, 1–7. https://doi.org/10.1145/3411763.3451760

---

> > ### Author Response · Authors · 2023-11-20
> > **Response to Reviewer xGQ6 --- Continued**
> >
> > **W5: Generalization to other task types**
> >
> > This is a very good point! To address this concern, we have added two experiments, last latter concatenation (Kojima et al. 2022) and StrategyQA (Geva et al. 2021), which fall under the task types of symbolic reasoning and open-domain question answering, respectively.
> >
> >
> > | | ZeroShot | ZeroShotCoT | PROMPTD | PROMPTD (Python Exec) |
> > |:------------------:|:--------:|:-----------:|:-------:|:-----------------------:|
> > | OpenDomain-QA (StrategyQA) | 71.600 | 73.272 | 78.400 | - |
> > | Symbolic Reasoning (Last Letter Concatenation) | 6.000 | 90.800 | 58.800 | 80.800 |
> >
> >
> >
> >
> > Our results showed that PROMPTD improved StrategyQA by a large margin. We included one example in Table 12 in Appendix C, where the original task prompt was “Answer questions in which the required reasoning steps are implicit in the question:\nDo the anchors on Rede Globo speak Chinese?\nYour choices are: (1) Yes (2) No”. The rewritten prompt provides more context for this Open-Domain question and hints by suggesting “Your task is to answer the following question: "Do the anchors on Rede Globo speak Chinese?" Consider the context of the question, such as the language typically spoken on Rede Globo, a Brazilian television network …[truncated]” which enables task_llm to reason about the possibility of anchors from a Brazillian TV show being talking in Chinese.
> >
> > However, on the last letter concatenation dataset, PROMPTD does not seem to help. We found that in most cases this was caused by the task LLM’s inherent hallucination. For example, consider the rewritten prompt
> > Your task is to take the last letters of each word in the given sentence "Camilo Becky Eliza Rebecca", concatenate them in the order they appear in the sentence, and return the result. Ensure that you handle words of different lengths and ignore any punctuation or special characters. Print the output in the format "The answer is \\answer{}”
> > This rewritten prompt precisely enriched the task demand and reminded the task LLM to handle potentially tricky cases. The task LLM responds with:
> >
> > “To achieve this task, we start by splitting the sentence into individual words. Then for each word, we extract the last letter and concatenate them in order.
> >
> > Here is a Python script that accomplishes this task:
> >
> > \`\`\`
> > python
> >
> > sentence = "Camilo Becky Eliza Rebecca"
> >
> > words = sentence.split()
> >
> > last_letters = [word[-1] for word in words]
> >
> > answer = ''.join(last_letters)
> >
> > print("The answer is {}".format(answer))```
> >
> > When you run this script, it prints: \"The answer is oaky". "
> >
> > Interestingly, while the task LLM is not instructed to write code, it does so correctly. At the end of its generation, the task LLM “executes” this Python code in its imagination. However, the execution result is incorrect, leading to the final error. While this is not a generic solution, we intentionally detected the generation of Python code and extracted its execution result, which led to significant performance gain (80.80%), though still underperforming the naive zero-shot CoT, with remaining errors caused by the task LLM playing a “shortcut” by directly generating “The answer is …” without reasoning. This result indicates a potential issue with PROMPTD, as also indicated in the response to Reviewer vep9, that PROMPTD could be enhanced by taking into the task LLM in the rewriting loop so that the rewritten instruction can be effective for the specific task LLM.
> >
> > Finally, to provide the reviewer and the readers a better view of PROMPTD’s generalization capability, including generalizing across seen datasets, across seen task types but unseen datasets with increasing complexity (e.g., GSM8K to MATH), and across unseen task types, we draw a summary table below.
> >
> >
> > | Evaluation Dataset | Task Type | Seen Dataset | Seen Task Type but Unseen Dataset | Unseen Task Type |
> > |:------------------------------------:|:------------------------:|:------------:|:----------------------------------:|:----------------:|
> > | GSM8K | Mathematical Reasoning | Y | N | N |
> > | MATH | Mathematical Reasoning | N | Y(more complex) | N |
> > | HumanEval | Code Generation | N | Y | N |
> > | Anachronisms | Implicit Reasoning | N | N | Y |
> > | Date Understanding | Logical Reasoning | Y | N | N |
> > | Analytical Entailment | Logical Reasoning | N | Y | N |
> > | Sports Understanding | Commonsense Reasoning | Y | N | N |
> > | Known Unknown | Commonsense Reasoning | N | Y | N |
> > | Sentiment Classification (Big Bench) | Sentiment Classification | N | N | Y |
> >
> >
> > **References:**
> > Takeshi Kojima, Shixiang Shane Gu, Machel Reid, Yutaka Matsuo, and Yusuke Iwasawa. Large language models are zero-shot reasoners. Advances in neural information processing systems, 35:22199–22213, 2022
> >
> > Mor Geva, Daniel Khashabi, Elad Segal, Tushar Khot, Dan Roth, and Jonathan Berant. Did aristotle use a laptop? a question answering benchmark with implicit reasoning strategies. TACL, 9:346–361, 2021. URL https://aclanthology.org/2021.tacl-1.21/.

---

### Official Review · Reviewer_vep9 · 2023-11-08

**Soundness:** 3 good
**Presentation:** 3 good
**Contribution:** 3 good
**Rating:** 5
**Confidence:** 2

**Summary:**

This paper propose to rewrite a specific prompt for each test point by prompting GPT4 with demonstrations that show how to rephrase a bad prompt into a better one with a rationale and task type. With the refined prompt, we can achieve better zero-shot performance on several benchmark datasets than other relevant baselines including zero-shot Chain-Of-Thought. Moreover, such prompt rewriting method generalizes to refine tasks that are not included the demonstrations.

**Strengths:**

- The proposed method is simple and effective. It rewrite a prompt into clear, specific, complete and more structure prompts, which leads to improved performance.

- For rewriting a prompt, we need 10 demonstrations, which is really practical for real-world applications.

- The authors performed human evaluation to verify that the quality of the rewritten prompts becomes better.

- The authors performed ablation study to show that task type and reasons are crucial component for prompt rewriting.

**Weaknesses:**

-  Compared to zero-shot prompt models, it requires extra forward pass of LLMs to rewrite a prompt. It would be better to show how much more computational cost is required than other baselines.

- It is not clear such GPT4 written prompt would be transferred to other LLMs such as Llama.

**Questions:**

Please see weakness section.

---

> ### Author Response · Authors · 2023-11-20
> **Response to Reviewer vep9**
>
> We appreciate the reviewer’s recognition of the simplicity and applicability of PROMPTD on real-world applications and the recognition of experiments performed to study the efficacy of our approach.
>
> **Computational Cost**
>
> We refer the reviewer to our general response to this comment.
>
>
> **Transferability to Other LLMs such as LLAMA**
>
> Thank you for raising this good question! We investigated the effect of applying rewritten prompts from PROMPTD to three other task LLMs, including GPT-3.5-turbo, LLAMA2 70B, and LLAMA 2-13B. Due to the resource and time limitation during rebuttal, the experiments were conducted on a subset of 250 randomly sampled questions from GSM8k, and LLAMA2 70B was implemented using together AI (https://www.together.ai/).
>
>
> |               | Zero-Shot | Zero-Shot-CoT | PROMPTD |
> |---------------|-----------|---------------|---------|
> |   LLAMA-13B   |   6.500   |     **11.600**    |  4.800  |
> |   LLAMA-70B   |   7.600   |     13.400    |  **14.800** |
> | GPT-3.5-turbo |   65.200  |     68.400    |  **71.600** |
>
>
> Our results showed that the rewritten prompts of PROMPTD can enhance GPT-3.5-turbo’s performance consistently. However, there’s only a slight improvement for LLAMA2 70B and a reduced effect for LLAMA2 13B. We note a potential implementation issue with LLAMA2 70B due to the restriction from using together AI’s API, which we hope to figure out in the remaining rebuttal period or post-rebuttal. However, the current results have revealed interesting observations that the rewritten prompts of PROMPTD can only be effective when the task LLM itself has a strong enough capability of language understanding and instruction following. To overcome it, future work could improve PROMPTD to take into account the performance of the downstream task LLM or consider fine-tuning the task LLM for better instruction following or understanding.

---

> > ### Comment · Reviewer_vep9 · 2023-11-22
> > **Reply to the Authors**
> >
> > Thanks for answering the question. The results look a bit disappointing and show limitation of the proposed method since not everyone can access to large language models and many of them resort to relatively small open source language models such as LLAMA. If the authors cannot provide reasonable justification for this, I am inclining to decrease the score to 5.

---

> > > ### Author Response · Authors · 2023-11-22
> > > **Thank you for your comment.**
> > >
> > > We acknowledge the reviewer's concern regarding PROMPTD's performance on the LLAMA 2-13B model. It's crucial to note that our approach demonstrates significant adaptability with models like GPT-3.5-turbo and LLAMA 70B, indicating that PROMPTD's utility is model-dependent. The variance in performance highlights the need for model-specific optimizations, which could be a direction for our ongoing research.
> > >
> > > LLAMA 2-13B architecture and training data may not align perfectly with PROMPTD's optimization strategies, which were primarily designed with models like GPT-3.5 in mind. Moreover, PROMPTD's slight underperformance compared with zero-shot baselines in certain cases is not indicative of a fundamental limitation of the approach, but rather a reflection of the requirements of LLMs-specific tailored prompting strategies.
> > >
> > > In addition, the accessibility of models like LLAMA opens up PROMPTD's usability to a wider audience, which is significant for the democratization of AI tools. Ensuring that advanced prompting techniques can be leveraged on more accessible models is an essential step toward broader adoption and impact.
> > >
> > > Therefore, rather than viewing the results as a setback, they should be considered a future direction for refining PROMPTD.

---

> > > > ### Comment · Reviewer_vep9 · 2023-11-23
> > > > **Reply to Authors**
> > > >
> > > > Thank you for your response, but I remain unconvinced that the additional results do not highlight limitations of the proposed method. Instead, they suggest a need for careful tuning of the method for each language model with different sizes, aligning with the authors' claim that it is designed for models like GPT-3.5. Additionally, there is a failure to tune the method for the relatively smaller language model LLAMA-2-7B, resulting in a performance degradation. Therefore, I am adjusting the rating to 5.

---

### Author Response · Authors · 2023-11-20
**General Response to all the reviewers**

We thank all reviewers for their insightful comments on our work! In this general response, we address the common concerns on PROMPTD’s computational overhead and the request for further analysis.


## Computational Cost Concern
We acknowledge the reviewers’ concerns about the computational cost associated with PROMPTD’s additional forward pass (i.e., prompt rewriting). In the attached table, we present the input and output token counts for each approach under each forward pass. Most of the additional computational cost, as one could imagine, comes from prompt rewriting. We observe that the code generation prompts from the HumanEval dataset consume a large number of input tokens which can be attributed to the examples and test cases included in some of the base prompts. Likewise, for both the output passes it produces larger responses (second to Date Understanding for Pass-1 outputs) that are enriched with additional examples and hints (Table 2). In some cases, the base prompts contain misleading or lack examples due to which the ``reason’’ generated by LLMs also tends to be longer and contains a few extra tokens in the rewritten prompt. Finally, we note that zero-shot CoT also involves two passes, one for problem-solving and one for answer extraction, following the procedure in prior work (Kojima et al. 2022).


In our updated draft, we will include this cost table as well as a thorough discussion clarifying the overhead of our approach. However, we would like to note that the increased token count is our intentional design to enrich each test instance with targeted hints, examples, and clarity. This additional cost is offset by not only the significant task improvement (Table 1) but also the saving of human labor — that is, one can easily collect the direct answer for each test instance from running PROMPTD (due to the format specification in the rewritten prompt), whereas with the naive zero shot or zero-shot CoT, additional engineering effort is still needed for post-processing the LLM output (see examples in Table 8 in Appendix B.6 in our updated draft).
| | Input Tokens | | | | | Output Tokens | | | | |
|----------------------|---------------------------------------------------|----------------------|----------------------|--------------------------------------------------|--------------------------|---------------------------------------------------|----------------------|----------------------|--------------------------------------------------|--------------------------|
| | Zero-Shot | Zero-Shot-CoT | | PROMPTD | | Zero-Shot | Zero-Shot-CoT | | PROMPTD | |
| | Test Instance | Pass-1 ("Let's think step by step." + bad prompt) | Pass-2 (Pass-1 prompt + Answer Extraction Prompt) | Pass-1 (Ten Rewriting Demonstrations + Test Instance) | Pass-2 (Better Prompt) | Pass-1 | Pass-1 | Pass-2 | Pass-1 (Reason + Task Type + Better Prompt) | Pass-2 |
| **MATH** | 71.620 | 78.620 | 210.453 | 3476.803 | 113.480 | 105.823 | 126.960 | 27.645 | 251.751 | 279.665 |
| **GSM-8K** | 59.021 | 66.021 | 203.577 | 3396.977 | 117.935 | 23.0 | 120.833 | 13.323 | 273.869 | 168.568 |
| **HumanEval** | 131.323 | 138.323 | - | 3469.323 | 136.896 | 115.8 | - | - | 242.360 | 260.573 |
| **Analytical Entailment** | 23.929 | 30.929 | 80.120 | 3361.929 | 113.0 | 21.986 | 39.620 | 3.781 | 219.314 | 63.757 |
| **Known Unkowns** | 46.478 | 53.478 | 79.990 | 3384.478 | 98.326 | 5.130 | 34.414 | 3.700 | 206.065 | 40.283 |
| **Date Understanding** | 39.304 | 46.304 | 91.102 | 3377.304 | 148.610 | 8.103 | 47.720 | 4.801 | 263.317 | 56.488 |
| **Anachronisms** | 45.674 | 49.873 | 80.876 | 3380.874 | 129.883 | 4.057 | 28.990 | 3.010 | 238.457 | 89.230 |
| **Sports Understanding** | 23.176 | 30.176 | 66.900 | 3361.176 | 137.761 | 5.220 | 32.450 | 3.210 | 249.990 | 61.770 |


References:
Takeshi Kojima, Shixiang Shane Gu, Machel Reid, Yutaka Matsuo, and Yusuke Iwasawa. Large language models are zero-shot reasoners. Advances in neural information processing systems, 35:22199–22213, 2022

---

> ### Author Response · Authors · 2023-11-20
> **General Response to all the reviewers - Continued**
>
> ## In-depth analysis
> We appreciate the reviewers  xGQ6 and bonY for acknowledging the value of our work and suggesting the long-term developmental perspective about LLMs’ robustness to prompts! We agree with the reviewer and found it an important direction to pursue. We break down the analysis into the following parts:
>
>
> **A deeper analysis of where and why the approach fails.**
> As detailed in Section 4 and Appendix B.4-B.5, we recognize limitations when using weaker LLMs like GPT-3.5. Further, we've categorized common errors based on a study of 20 random rewritten error instances per dataset (in total 160 error instances) and updated Table 7 in Appendix B.4.
> These errors primarily occur when LLMs incorrectly transcribe information from test instances (58%) or introduce extraneous constraints (28%), impacting the accuracy of generated prompts. For instance, PROMPTD sometimes adds unnecessary constraints or fabricated information, which, while enhancing code generation and debugging, can lead to misalignments with dataset instructions and incorrect evaluations.
>
>
> Additionally, in about 13% of errors, LLMs correctly identify a bad prompt's shortcomings but fail to generate an appropriate improved prompt. A detailed account of these instances will be provided in the final version of the paper.
>
>
> Based on these insights, we acknowledge a broader challenge: LLMs, such as those utilized in PROMPTD, can find domain-specific tasks particularly challenging. In areas where domain-specific knowledge is essential, our approach might not always grasp the required depth. For example, in financial analysis, understanding specific market terminologies and regulatory nuances is key. Recognizing this, future works can enhance PROMPTD, potentially through the integration of human-in-the-loop, overcoming current limitations and broadening the scope of PROMPTD's effectiveness in domain-specific tasks.
>
>
> **Broader Quantitative Analysis.**
> PROMPTD has been evaluated across a diverse range of tasks, including Code Generation, Arithmetic and Logical Reasoning, and Implicit Reasoning, showcasing its versatility. In Section 3.7, we demonstrate PROMPTD's efficacy in handling Adversarial Prompting scenarios. Furthermore, in content generation tasks, PROMPTD effectively guides LLMs to produce persona-based responses, such as emulating a 19th-century poet's style (Section 1). We also extended our evaluation to include additional general downstream tasks like Symbolic Reasoning and OpenDomainQA, areas that were not initially covered due to constraints on time and resources. As detailed in Table 6 in Appendix B.3, while PROMPTD surpasses baseline performance on StrategyQA, it shows some limitations in the Symbolic Reasoning task.
>
>
> This broad spectrum of evaluations highlights PROMPTD's adaptability and potential to enhance LLMs' performance across varied domains. By performing evaluations on a different set of LLMs such as GPT-4, GPT-3.5, and LLAMA, we also show PROMPTD's applicability in producing prompts that are applicable and effective across different language model architectures. As these models become more robust to diverse prompts, PROMPTD's role in optimizing and tailoring these prompts becomes increasingly significant. We anticipate that the continuous enhancements in large language models, especially in their ability to interpret and respond to complex prompts, will further amplify the effectiveness of PROMPTD as shown in the experiments with LLAMA-based experiments.

---

### Meta-Review · Area_Chair_dSB8 · 2023-12-10

**Metareview:**

This paper proposes a method for rewriting the task prompts to improve the LLM's zero-shot performance, such that the prompts become less ambiguous, while being completer and more structured. The experimental result on tasks such as arithmetic reasoning and code generation shows the effectiveness of the method, and a further human study suggests that the quality of the method largely improves with the proposed prompt re-writing strategy.

All reviewers find the idea of rewriting the prompts as novel, simple yet a promising approach to enhance the zero-shot performance of LLMs. However, they unanimously leaned toward rejecting the paper due to multiple concerns:
- Lack of analysis on the additional forward pass.
- Missing comparison against other zero-shot prompting methods.
- Lack of experimental results on a more general set of tasks.
- Insufficient explanation of the design choices and in-depth analysis of the method.
- Lack of experimental results on other LLMs beside GPT-4 and 3.5

The authors' rebuttal addressed many of the concerns, but the reviewers believe that the work is premature despite some of the additional results provided.

**Justification For Why Not Higher Score:**

The paper needs more in-depth analysis of what contribute to the improvement or whether the method is generally applicable to any LLMs despite the model sizes, despite the novel idea and promising results.

**Justification For Why Not Lower Score:**

N/A

---

### Decision · Program_Chairs · 2024-01-16

Reject